# Conserved biophysical compatibility among the highly variable germline-encoded regions shapes TCR-MHC interactions

**Christopher T Boughter\*, Martin Meier-Schellersheim\***

Computational Biology Section, Laboratory of Immune System Biology, National Institute of Allergy and Infectious Diseases, National Institutes of Health, Bethesda, United States

**Abstract** T cells are critically important components of the adaptive immune system primarily responsible for identifying and responding to pathogenic challenges. This recognition of pathogens is driven by the interaction between membrane-bound T cell receptors (TCRs) and antigenic peptides presented on major histocompatibility complex (MHC) molecules. The formation of the TCR-peptide-MHC complex (TCR-pMHC) involves interactions among germline-encoded and hypervariable amino acids. Germline-encoded and hypervariable regions can form contacts critical for complex formation, but only interactions between germline-encoded contacts are likely to be shared across many of all the possible productive TCR-pMHC complexes. Despite this, experimental investigation of these interactions have focused on only a small fraction of the possible interaction space. To address this, we analyzed every possible germline-encoded TCR-MHC contact in humans, thereby generating the first comprehensive characterization of these largely antigen-independent interactions. Our computational analysis suggests that germline-encoded TCR-MHC interactions that are conserved at the sequence level are rare due to the high amino acid diversity of the TCR CDR1 and CDR2 loops, and that such conservation is unlikely to dominate the dynamic protein-protein binding interface. Instead, we propose that binding properties such as the docking orientation are defined by regions of biophysical compatibility between these loops and the MHC surface.

**\*For correspondence:**
christopher.boughter@nih.gov
(CTB);
mms@niaid.nih.gov (MM-S)

**Competing interest:** The authors declare that no competing interests exist.

## Editor's evaluation

This important work presents evidence that evolved biophysical compatibility between T cell receptors (TCRs) and MHC molecules is possible and a potential solution to the question of how TCRs could be biased towards MHC proteins given the massive diversity in both receptor and ligand. The evidence supporting the claims of the authors is solid, although the nature of these evolutionary questions makes it difficult to confidently answer some of the raised questions. The work will be of interest to immunologists, structural biologists, and evolutionary biologists.

## Introduction

The interaction between T cell receptors (TCRs) and the peptide-presenting major histocompatibility complex (pMHC) molecule is a nanometer-scale recognition event with macroscopic consequences for health and control of disease. In αβ T cells, the six complementarity-determining region (CDR) loops of the TCR are responsible for scanning the pMHC surface and determining the appropriate immunological response. Since the first crystal structures of a TCR-pMHC complex were solved in

1996 (*Garcia et al., 1996*; *Garboczi et al., 1996*), general 'rules of engagement' between TCR and pMHC have been identified and reliably reproduced in subsequent structures (*Garcia et al., 1999*; *Yin et al., 2012*). First, the peptide is presented in the 'groove' formed by the two α-helices of either the class I or class II MHC molecule (*Bjorkman et al., 1987*; *Brown et al., 1993*). The TCR then scans this peptide primarily using the highly diverse CDR3 loops of the TCRα and TCRβ chains, whose sequences are determined by V(D)J recombination (*Wu et al., 2002*; *Jung and Alt, 2004*). The sequences of the remaining two CDR1 and CDR2 loops of the α and β chains are entirely determined by germline TRAV and TRBV genes, respectively. In a majority of structures, these CDR1 and CDR2 loops make a number of key contacts with well-defined regions of the likewise germline-encoded MHC $\alpha$-helices (*Gras et al., 2012*).

Importantly, previous work has found a large list of deviations from these classic 'rules of engagement'. The hypervariable CDR3 residues can dominate the interactions with the germline-encoded MHC α-helices (*Piepenbrink et al., 2013*; *Singh et al., 2022*), germline-encoded CDRs can interact strongly with peptide (*Piepenbrink et al., 2013*), an exceptionally long peptide can 'bulge' causing separation between germline-encoded regions (*Tynan et al., 2005*), or the docking orientation can even be entirely reversed (*Beringer et al., 2015*; *Gras et al., 2016*; *Zareie et al., 2021*). When considering the large number of possible TCR-pMHC interactions and the substantial crossreactivity of TCRs (*Riley et al., 2018*; *Sewell, 2012*), these historically non-canonical interactions may in fact be rather common. Despite this great variability inherent in TCR-pMHC complex formation, the docking orientation of the TCR over MHC molecules (*Figure 1A and B*) falls within a tight distribution of angles for the majority of structures solved thus far (*Figure 1C and D*) nearly independent of the strong variation in the CDR3 and peptide sequences involved. This suggests that interactions between germline-encoded regions play fundamentally important roles.

However, it remains unclear whether germline-encoded residues enforce the orientational preference highlighted in *Figure 1*, or if this preference is a consequence of thymic selection (*Van Laethem et al., 2007*; *Tikhonova et al., 2012*; *Van Laethem et al., 2012*), whereby T cells are first positively selected on TCR recognition of MHC molecules and later negatively selected against autoreactivity towards MHC molecules loaded with organismal self-peptides (*Blackman et al., 1990*; *Juang et al., 2010*; *Germain, 2002*). This selection occurs in the presence of the potentially orientation-determining co-receptors CD4 or CD8. However, evidence for this hypothesis of docking determined by selection comes largely from a study utilizing a knockout system that deviates strongly from the physiological norm (*Van Laethem et al., 2007*), or from the identification of rare 'reverse-docking' TCRs (*Beringer et al., 2015*; *Gras et al., 2016*; *Zareie et al., 2021*). Alternatively, the extent of this orientational preference may reflect a co-evolution of interacting, germline-encoded regions of TCR and MHC molecules. In support of this hypothesis, a number of mouse studies have identified key residue pairs critical for TCR-pMHC binding which were later structurally validated and found in other, distantly related organisms (*Scott-Browne et al., 2009*; *Krovi et al., 2019*; *Feng et al., 2007*; *Blevins et al., 2016*; *Scott-Browne et al., 2011*; *Dai et al., 2008*). While these studies find evidence of conserved interactions for a small subset of alleles, no work thus far has found consistent conservation across the full range of possible TCR-MHC molecule interactions.

These observations underscore the need for a more thorough understanding of the fundamental role of CDR1 and CDR2 in the recognition of pMHC complexes, either as sculptors of the TCR-pMHC interface, or as opportunistic reinforcements to a protein-protein interaction largely determined by the CDR3 loops and peptide involved. Due in part to the concentration of diversity at the CDR3-peptide interface, a majority of studies characterizing TCR-pMHC interactions have focused on describing immune recognition events in the context of this interface (*Birnbaum et al., 2014*; *Sharon et al., 2016*; *Sibener et al., 2018*; *Krovi et al., 2019*; *Burrows et al., 2010*; *Ishigaki et al., 2022*). This runs counter to the relative contributions of the germline-encoded regions that account for a majority (75–80%) of the binding interface (*Garcia et al., 2009*). The progress made thus far can largely be attributed to X-ray crystallography, the gold standard for the study of TCR-pMHC interactions. However, this approach is severely limited by its low-throughput nature, calling its utility for this particular combinatorial problem into question. While a majority of the 45 productive TCRα-encoding TRAV and 48 productive TCRβ-encoding TRBV genes in humans have been crystallized at least once in complex with pMHC (*Gowthaman and Pierce, 2019*), the total of 149 crystallized human TCR-pMHC complexes pales in comparison to the 2160 possible productive TRAV-TRBV pairings. When

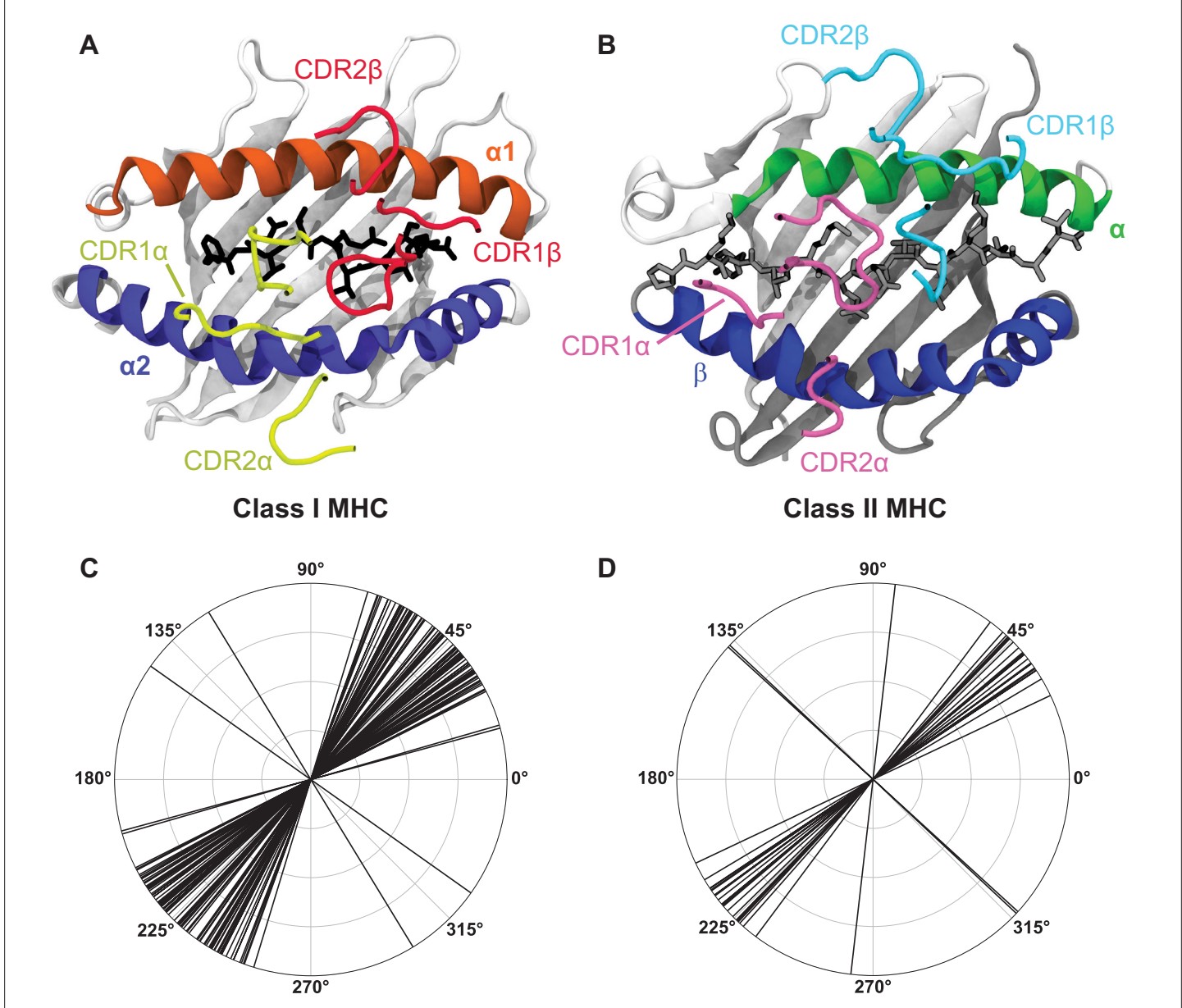

**Figure 1.** A breakdown of the canonical docking orientation adopted by TCRs over the pMHC complex in all solved crystal structures to date highlights the strong structural conservation of the TCR-pMHC interaction. (**A, B**) Example renderings of TCR-pMHC complexes for class I (A, PDB: 6MTM) and class II (B, PDB: 1J8H) MHC molecules. The CDR loops are largely representative of the placement over the MHC helices and peptide for the vast majority of TCR-pMHC complexes. (**C, D**) Polar coordinate plots of TCR docking angles over class I (**C**) and class II (**D**) MHC molecules (data via the TCR3D database *Gowthaman and Pierce, 2019*). Note that these docking angles do not perfectly overlay on to the structures of panels A and B, as there are also slight deviations in the location of the TCR center of mass over the MHC complex.

we further consider the nearly 900 known unique amino acid sequences of class I MHC in humans (also referred to as class I human leukocyte antigen - HLA), we find that structural studies to date have covered less than 0.01% of the total possible TCR-MHC germline interaction space.

While structural modeling is an intriguing alternative (*Milighetti et al., 2021*), predictions of the conformations adopted by flexible TCR or antibody CDR loops and the placement of the antigen-contacting amino acid side chains on these loops have been found to be somewhat unreliable (*Evans et al., 2022*). In the work reported here, we explore how much insight about TCR-MHC interactions can be gained using a simplified, pseudo-structural representation of the biophysics of protein inter-actions that scores amino acid interactions as either productive, neutral, or counter-productive with

regard to complex formation. Using the Automated Immune Molecule Separator (AIMS) software (**Boughter et al., 2020**; **Boughter and Meier-Schellersheim, 2023**) we generated the first systematic characterization of every germline-encoded TRAV-MHC and TRBV-MHC interaction through such a pseudo-structural approach. The AIMS architecture allows us to analyze the CDR1 and CDR2 contributions to interactions with MHC molecules independent of CDR3, providing a not previously explored antigen-independent perspective on the TCR-MHC complex. We find that the great diversity present in the CDR1 and CDR2 loops encoded by TRAV and TRBV genes in humans, as well as the dynamic nature of amino acid side chains, makes strongly conserved interactions between TCR germline-encoded residues and MHC $\alpha$-helices highly unlikely. This suggests that the few strongly evolutionarily conserved interactions that have been identified previously in the literature should be considered exceptions and not the rule. Importantly, however, analyzing the biophysical compatibility for each germline-encoded CDR residue and those on the MHC α-helices, we find evidence of an evolutionary conserved region of permissible binding on the MHC molecular surface responsible for the observed canonical orientation of the TCR-MHC molecular complex. These measures of biophysical compatibility between germline CDRs and MHC α-helices can be used as a basis for understanding TCR-pMHC interactions, and as a tool for understanding how perturbations to these regions alter these interactions.

## Results

Starting from the TRAV and TRBV protein entries from the ImMunoGeneTics (IMGT) database (**Brochet et al., 2008**; **Lefranc, 2014**) and the IMGT-HLA database containing all identified HLA alleles (**Mack et al., 2013**), we include only productive genes and unique amino acid sequences in our final analysis dataset. This final dataset included a total of 882 unique class I MHC sequences (HLA-A, B, and C), 44 class II α chain sequences (HLA-DQα, DRα, and DPα), and 431 class II β chain sequences (HLA-DQβ, DRβ, and DPβ). Due to significant differences in the alignments and more nuanced differences in the structures of these MHC molecules, we analyzed these three groups separately and compare the results as appropriate.

### Polymorphic HLA molecules are far less diverse than human TRAV and TRBV genes

We first searched for potentially conserved interactions between TCR and MHC molecules that would suggest a strong coevolution between the two. Such biases in the amino acid usage of TCR and MHC molecules at specific sites can occur either in the form of full amino acid conservation, or as pairs of meaningfully co-varying amino acids found across multiple diverse sites. Without the structural data necessary to precisely identify these conserved interactions, we can characterize the diversity present in TRAV, TRBV, and HLA amino acid sequences using bioinformatic approaches and determine whether a conserved germline-encoded interaction is possible for each unique molecular combination. We quantified the diversity in these molecular subsets using concepts from information theory. Specifically, the amino acid diversity of each TRAV, TRBV, and HLA sequence encoded into an alignment matrix (**Figure 2A**) was quantified by calculating the Shannon entropy (**Shannon, 1948**) for each individual dataset (**Figure 2B**). The alignment matrix of **Figure 2A** provides the basis for all of our pseudo-structural analyses throughout this manuscript. The incorporated structural information goes as far as picking out the CDR loops of each TRAV/TRBV sequence and $\alpha$-helices of each MHC sequence, but does not make explicit assumptions about relative structural orientations.

   **Figure 2B** indicates that the TRAV and TRBV genes are very diverse, even more so than the polymorphic HLA alleles. Despite the low number of TRAV and TRBV genes significant diversity persists in CDR1 and CDR2, suggesting limited conservation in those residues most likely to contact the MHC surface. Further, most of the entropy in MHC class I alleles is concentrated in the peptide-binding region. In this more diverse region the entropy tops out at around 2 bits, which corresponds to at most 4 equiprobable amino acids, or a distribution of a slightly larger number of amino acids strongly skewed towards one particular amino acid. In the α-helical regions that are in contact with the germline-encoded CDR loops, the class I entropy reaches a maximum of 1.5 bits, or at most 3 equiprobable amino acids. These same trends persist for the analysis of class II HLA alleles, with

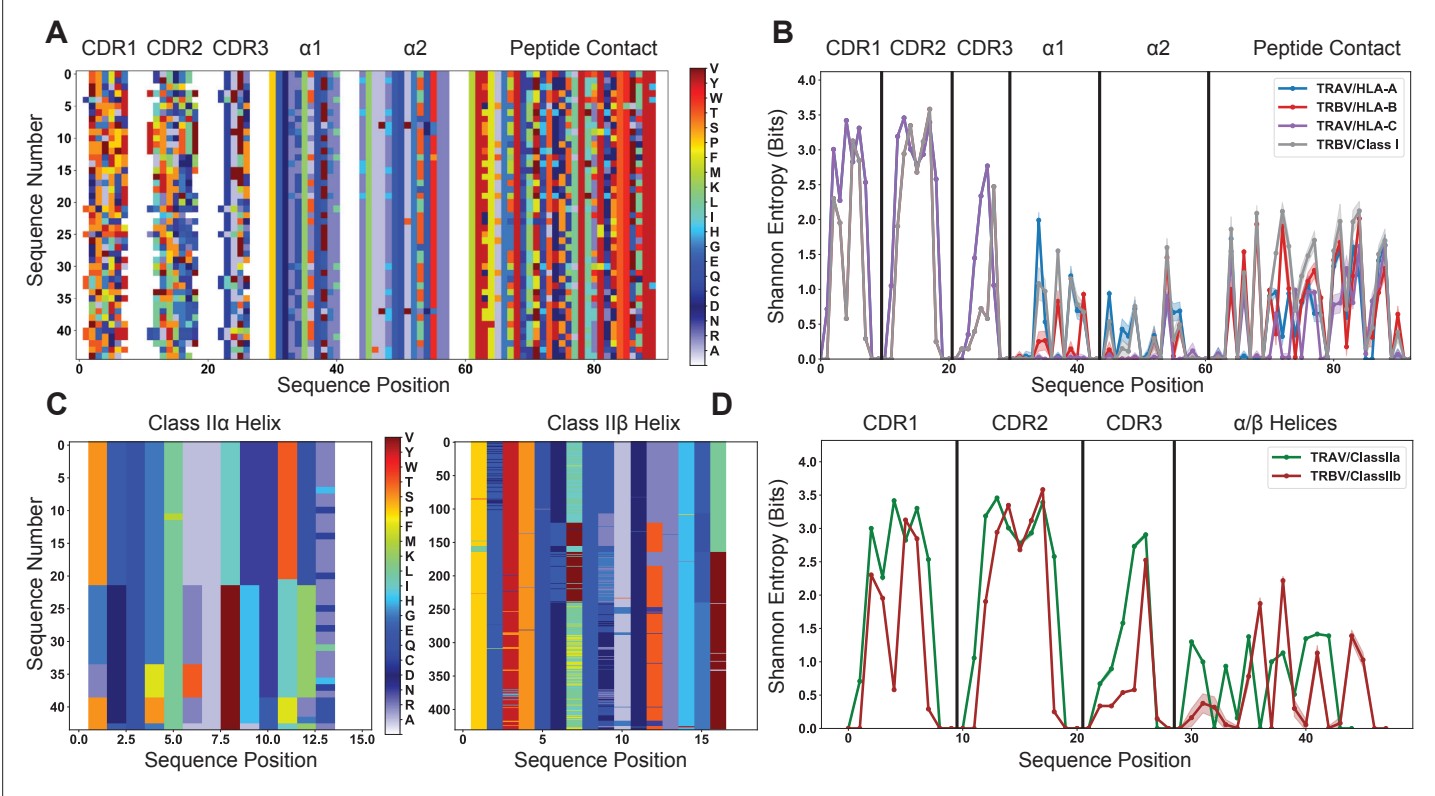

**Figure 2.** Quantification of diversity of TRAV, TRBV, and HLA alleles reveals limited sites for fully conserved interactions. (**A**) Visualization of the TRAV amino acid sequences paired with a subsampled HLA class I dataset. (**B**) This subsampling is repeated 1000 times to generate the average population Shannon entropy for each allelic subset as a function of position in the matrix of (**A**). (**C**) Visualization of the alignment-encoded matrices of HLA class IIa and classIIb datasets. (**D**) Calculated position-sensitive entropy for the HLA class IIa and class IIb datasets using the same subsampling scheme as (**B**). The color bar for panels (**A**) and (**C**) give the amino acid key for the matrix encoding, where each individual color represents an individual amino acid, and gaps in the matrix are colored white. In panels (**B**) and (**D**), the maximal entropy for the 20 possible amino acids at a given site is 4.3 bits, whereas an entropy of 0 bits represents a fully conserved amino acid. Variation in the entropies over subsampling repetitions is represented by the standard deviation as shadowed regions around the solid line averages.

The online version of this article includes the following figure supplement(s) for figure 2:

**Figure supplement 1.** Quantification of mutual information (MI) between germline encoded TCR regions and MHC α-helices show no strong signal.

**Figure supplement 2.** Information theoretic analysis of TCR-MHC pairs from a range of mammalian species shows that strongly co-varying residues between TCR and MHC germline sequences are rare.

**Figure supplement 3.** Mutual information differences between calculated TRAV and TRBV quantities from *Figure 2—figure supplement 2C, D* (**A**), or from class IIa (**B**), or class IIb (**C**) quantities.

**Figure supplement 4.** Information theoretic metrics for crystallized TCR-MHC pairs validate the combinatorial calculation approaches of *Figure 2* and *Figure 2—figure supplements 1–3*.

---

sequence diversity more limited in the class II α-helices when compared to the germline-encoded CDR loops (*Figure 2C and D*).

HLA alleles are polymorphic, which can be misconstrued as having high sequence diversity in MHC molecules across the human population. However, the polymorphic sites are limited to a minority of the total variable positions within the MHC nucleotide sequences (*Robinson et al., 2017*), which suggests even more limited variability at the amino acid level. While there are many unique alleles, these alleles are largely similar, especially when compared to other, more diverse protein families that are strongly structurally conserved but sequentially divergent, as in the neuronal Dpr-DIP system in *Drosophila melanogaster* (*Nandigrami et al., 2022*). Conversely, despite the low number of TCR germline genes present in humans, the diversity is substantial, with multiple sites across the CDR loops harboring over 10 unique amino acids. This mismatched diversity between the TCR germline-encoded CDR loops and the MHC α-helices suggests that conserved germline-encoded interactions

are unlikely to exist for every possible molecular combination, either within a single individual or across the human population more broadly. Mutual information calculations (see Methods) corroborate these observations by quantifying this lack of co-variation of germline-encoded TCR and MHC residues in both humans (*Figure 2—figure supplement 1*) and in a wider range of mammals (*Figure 2—figure supplements 2 and 3*). If we limit these mutual information calculations solely to those TCR-MHC pairs that have been crystallized in complex, we see this lack of co-variation persists (*Figure 2—figure supplement 4*).

## Biophysical diversity in TRAV and TRBV genes makes strongly conserved contacts across HLA molecules unlikely

The diversity in TRAV and TRBV germline-encoded regions that we had found excluded the possibility of conserved contacts as key mediators across all TCR-MHC germline-encoded interactions. However, this did not rule out the possibility that within this diversity, there are conserved biophysical interactions. For example, a site on CDR2β could be highly diverse, but only allow expression of hydrophilic residues of neutral charge. Pairing this with an α-helical site on HLA molecules which likewise only permits hydrophilic residues would represent a likely site of conserved interaction. We can test for such relations by quantifying the position-sensitive amino acid biophysical properties for both the germline-encoded TCR (*Figure 3A*) and HLA (*Figure 3B*) residues. Looking at both the position-sensitive charge (*Figure 3*) and the position-sensitive hydropathy (*Figure 3—figure supplement 1*), we see that the substantial diversity in the TRAV and TRBV sequences corresponds to a similar biophysical diversity in the amino acids found across the germline-encoded CDR loops.

To put these results into perspective, we compared them to corresponding variabilities found for another class of molecules that interact with the MHC. The killer cell immunoglobulin-like receptors (KIRs) largely recognize the MHC class I subset HLA-C, subsequently relaying either inhibitory or stimulatory signals to the natural killer cell expressing these KIRs (*Thielens et al., 2012*). While each individual possesses a fixed subset of KIRs, with no analogous diversity introduced by a V(D)J-like process, polymorphisms in these KIRs exist across the human population (*Robinson et al., 2010*). The interactions between these KIRs and HLA-C molecules are more well-defined than the germline-encoded interactions between TCR and MHC molecules, with strong salt bridge and hydrogen bond networks in place between two immunoglobulin-like domains of the KIR and each α-helix of HLA-C (*Figure 3—figure supplement 2A, B*; *Moradi et al., 2015*; *Moradi et al., 2021*).

We found that from an initial dataset of 369 KIR polymorphisms compiled from the IPD-KIR database (*Robinson et al., 2010*), only 31 productive alleles have unique amino acid sequences in the MHC molecule-binding region with very limited diversity (*Figure 3—figure supplement 2C, D*). Further, within the few moderately diverse sites, we found almost no variation in the biophysical properties of the amino acids contacting the MHC molecule, or in those of the amino acids on the HLA-C α-helices (*Figure 3C and D*). We further contextualized these findings with specific examples from KIR-MHC and TCR-MHC molecular interactions (*Figure 3E and F*). We selected a key hydrogen bonding network in the KIR2DL2-HLA-C*07:02 interface (*Moradi et al., 2021*) and compared this to the evolutionarily conserved YXY motif of CDR2β (*Feng et al., 2007*; *Scott-Browne et al., 2011*) found in a small subset of human TRBV genes. From this, we can see how diversity in each region reinforces or disrupts potentially conserved interactions at each site. Across all KIRs tested, the interacting residues are fully conserved, suggesting diversity is not tolerated at these sites. Similarly, diversity is exceptionally limited at the interacting sites on the HLA-C α2 helix.

Contrasting this with the regions involved in the YXY motif interactions, we see much higher diversity across both the TCR and MHC molecules. Considering specifically the conserved lysine on the α1 helix, we see that some of the residues utilized by TRBV-encoded CDR2 would create a strong conflict with the proposed conserved interaction site. While the YXY motif present on a few TRBV genes is clearly evolutionarily conserved (*Scott-Browne et al., 2011*) it is evident that substitutions either on CDR2β or on the MHC α1 helix would disrupt this interaction. These results strongly suggest that conserved interactions between germline-encoded CDR loops and MHC α-helices are deviations from the norm, and that CDR1 and CDR2 loops of TCRα and TCRβ likely adopt a more opportunistic binding strategy.

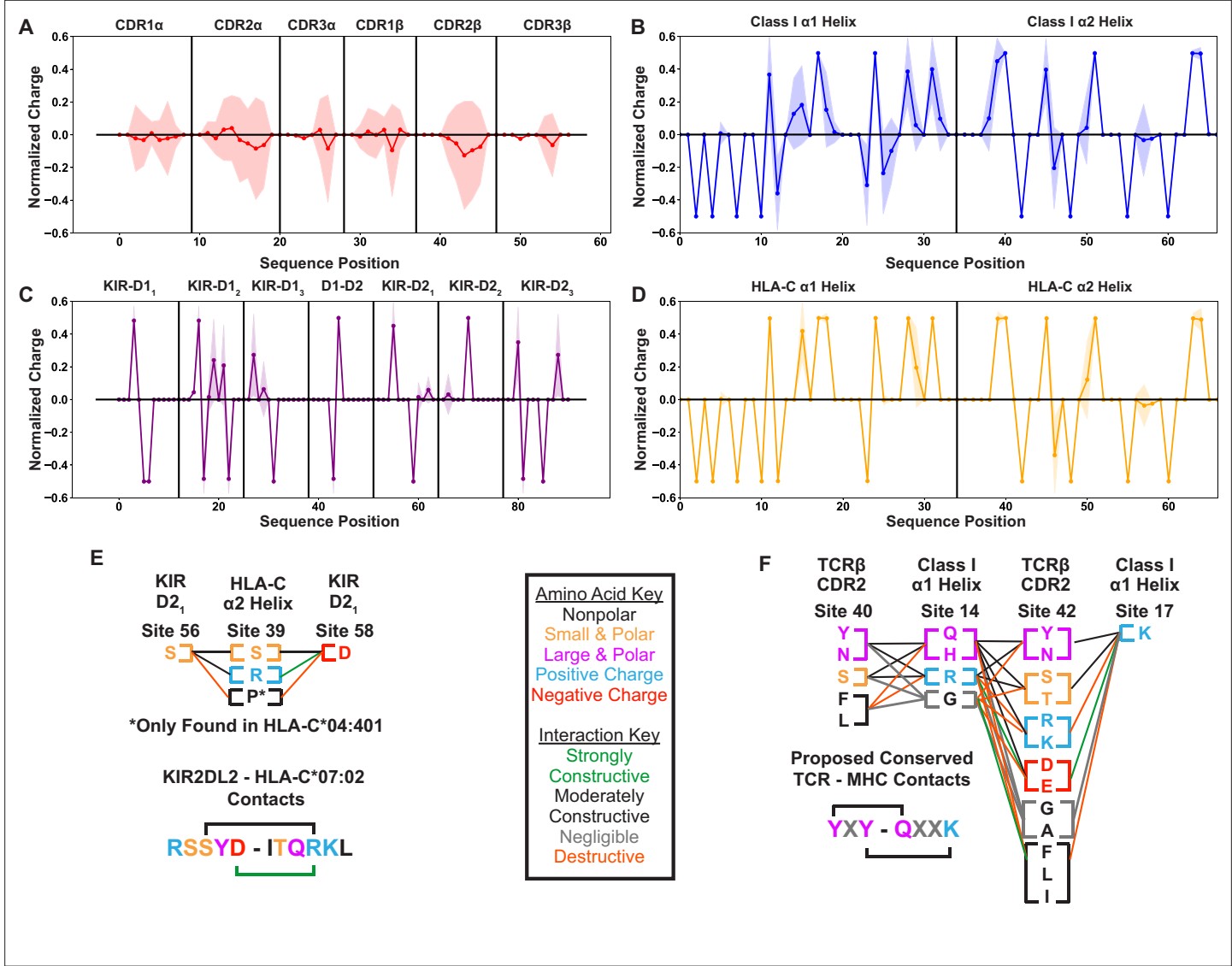

**Figure 3.** Position sensitive biophysical characterization of germline-encoded CDR loops highlights increased variation when compared to KIR or MHC sequences. Position sensitive amino acid charge averaged over all germline-encoded TCR CDR loops (n=48) (**A**), MHC class I α-helices (n=882) (**B**), KIR MHC-contacting regions (n=31) (**C**), or HLA-C α-helices (n=265) (**D**). Solid lines represent averages over unique amino acid sequences, while the standard deviations about these averages are given by the shaded regions. Visualization of specific interactions for KIR-MHC (**E**) or TCR-MHC (**F**) and their relative conservation across each respective dataset. Interactions are represented either in a single sequence context where a known binding network exists or as columns of residues present across polymorphisms or across TRBV genes. Key in the center gives color coding for either the biophysical properties of each amino acid (letters) or the relative contribution of these amino acids to an interaction interface (lines). Charges are normalized to a mean of 0 and a standard deviation of 1. Only polymorphisms across two-domain KIRs with well-characterized binding partners (KIR2DL1, 2DL2, 2DL3, 2DS1, 2DS2) are considered.

The online version of this article includes the following figure supplement(s) for figure 3:

**Figure supplement 1.** Position-sensitive hydropathy shows similar trends to the position-sensitive charge, showing higher variability in the germline-encoded TCR CDR loops than all other tested datasets.

**Figure supplement 2.** The killer cell immunoglobulin-like receptors (KIRs) and their recognition of HLA-C represent a suitable comparison to the germline-encoded CDR loops and their recognition of MHC.

## Calculation of TCR-MHC interaction potentials finds strong differences in germline-encoded recognition strategies

While our information-theoretic and biophysical analyses suggested limited conserved contacts between TCR α/β CDR1 and CDR2 loops and MHC helices, they did not rule out coevolution at the

level of biophysical compatibility over a broader interaction region. To explore the possibility of such coevolution that does not manifest itself at the level of genetic correlations, we analyzed TCR-MHC compatibility using the previously-validated (*Nandigrami et al., 2022*) AIMS interaction potential. Using a simplified potential to estimate protein interaction propensities, analyses performed with AIMS contain few implicit assumptions and compare favorably with more detailed and computationally expensive models. In a binary classification of a large database of structurally similar protein complexes, a combination of the AIMS interaction potential and a linear discriminant analysis-based classifier was capable of distinguishing binders and non-binders to an accuracy of 80%, whereas calculations run on over 45 µs of simulated all-atom trajectories could only distinguish to an accuracy of 50%. While not as biophysically descriptive as methods such as alchemical free energy perturbation (*Gumbart et al., 2013b*) or potential of mean force-based calculations (*Gumbart et al., 2013a*), the AIMS interaction potential generates accurate predictions for problems that are intractable with current limitations to computation due to the number of protein complexes of interest.

These interaction scores are calculated between all possible interacting amino acid sequences of HLA alleles and TRAV or TRBV genes (i.e. without preference for a canonical binding orientation), providing a useful quantification of the relative contribution of germline-encoded CDR1 and CDR2 loops to a given TCR-MHC complex (See Methods). *Figure 4* shows the results of this scoring metric, focusing on the resultant patterns evident in both the x-axis, corresponding to the TRBV (*Figure 4A*) or TRAV (*Figure 4B*) genes, and the y-axis, corresponding to each HLA allele. We note that the variance in the strength of interaction with MHC class I is greater for the CDR1 and CDR2 loops of TCRα than those of TCRβ (*Figure 4—figure supplement 1*). Our analysis showed that TRAV38-1, 38–2, and 19 have the strongest potential interactions with class I HLA molecules of any TCR genes. Conversely, the TRBV genes 7–2 and 7–3 are predicted to have a severely limited interaction potential with HLA molecules (*Figure 4C and D*). Interestingly, these alleles, TRBV7-2 and 7–3, are of significant interest in autoimmunity as they have been associated with celiac disease (*Gunnarsen et al., 2017*; *Qiao et al., 2014*) and multiple sclerosis (*Sethi et al., 2013*), respectively. Similarly, those TRBV alleles most strongly enriched in celiac patients (7–2, 20–1, and 29–1) (*Qiao et al., 2014*) likewise all have a corresponding low AIMS interaction potential with class I and class II HLA molecules (*Figure 4—figure supplement 2*).

These interaction potentials suggest that not only do germline-encoded CDR loops not contact the same conserved regions of the HLA molecule, but further imply that different TRAV/TRBV gene usage encourages differential utilization of the CDR loops. For instance, TRBV6-5 has a high CDR1 interaction potential and a negligible CDR2 interaction potential. Conversely, the suggested CDR bias is reversed for TRBV11-2. These trends are similarly found for analysis of germline-encoded interactions with HLA class II molecules (*Figure 4—figure supplement 2*). These results further highlight the variability across the full repertoire of germline-encoded CDR1 and CDR2 loops and the role of this variability in interactions with MHC molecules.

## Interaction scores show good agreement with structural data

To test our scoring of CDR-MHC interactions, we analyzed all TCR-pMHC complexes crystallized thus far. Using the TCR3D Database (*Gowthaman and Pierce, 2019*), we were able to process 182 total human TCR-pMHC complexes and quantify germline-encoded contacts. Final interactions included in the analysis were defined using somewhat generous cutoffs of 6 Å for charged interactions and 4.5 Å for van der Waals interactions (Methods). This comparison shows good agreement between our bioinformatic results and structural analyses (*Figure 5*). The results show that a normalized version of our AIMS interaction potential (*Figure 5A*) is in qualitative agreement with a symmetrized version of the count matrix generated from crystal structure contacts between TCR and class I MHC molecule side chains (*Figure 5B*). Normalization of these matrices is necessary, as residues with a negative or zero interaction potential are indistinguishable in the empirical contact map; that is, a negative number of contacts are not possible. Non-normalized versions of these matrices can be found in *Figure 5—figure supplement 1A, B*. The largest discrepancies between the interaction potential and the empirical matrix occurs in regions of hydrophobic interactions, as strongly hydrophobic residues are relatively rare on the MHC α-helices and TCR CDR loops (*Figure 5—figure supplement 1C*). Further, biases in the PDB from repeated crystallization of identical molecular species are evident, particularly in Ala-Tyr and Gln-Tyr MHC-TCR interactions (*Figure 5—figure supplement 1D*).

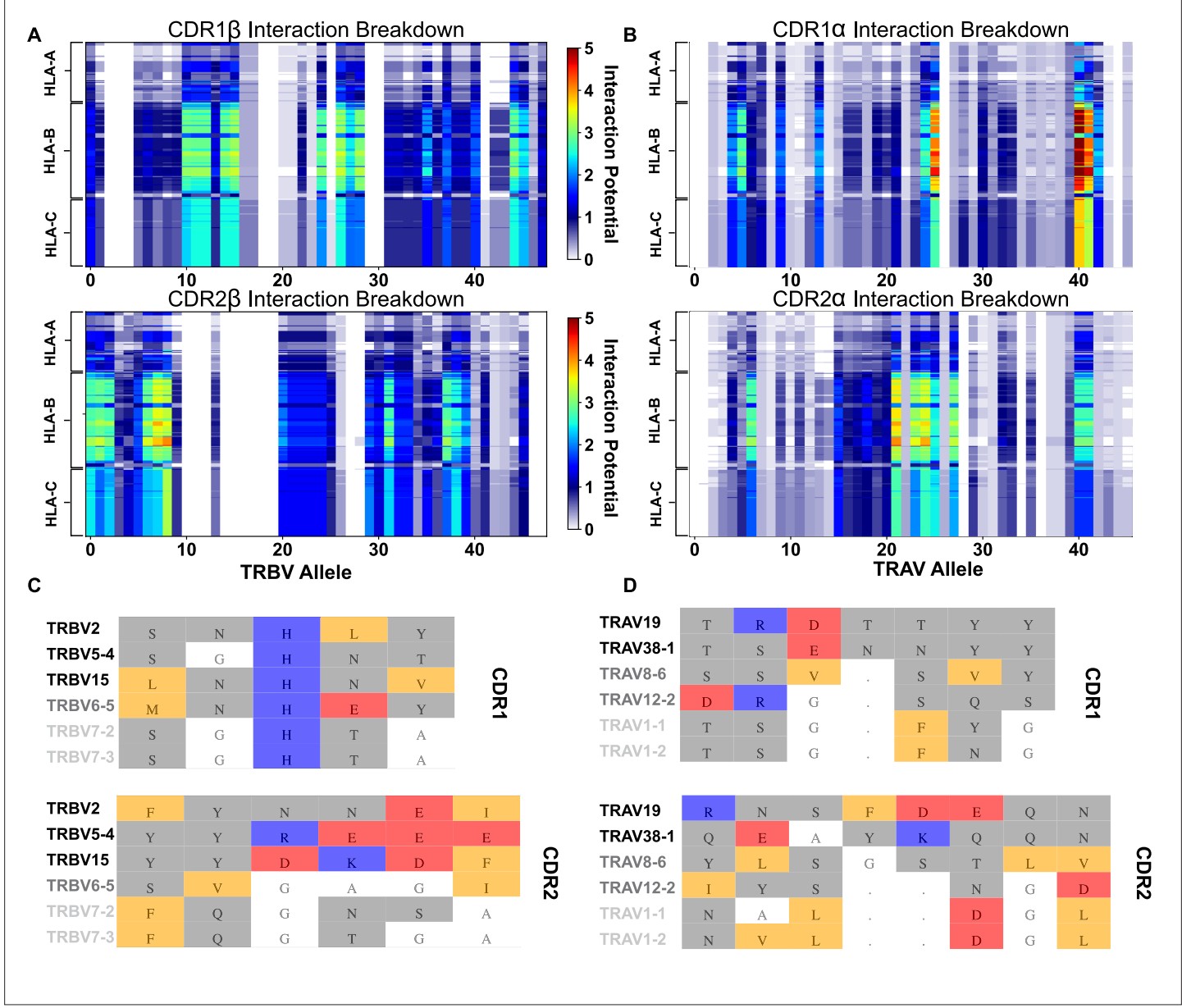

**Figure 4.** Interaction score between every TRBV (**A**) or TRAV (**B**) sequence and HLA allele for all four germline-encoded CDR loops. The x-axis moves across each productive germline-encoded TCR gene (see *Supplementary file 2* for key), while the y-axis is grouped by each broad HLA class I allele group. The color bar gives the interaction potential (unitless, higher potentials suggest stronger interactions). Alignments of TRBV (**C**) or TRAV (**D**) sequences highlighting genes with a range of interaction potentials, colored by biophysical property. Gene names for each sequence are colored by interaction potential: high-black, moderate-gray, and negligible-light gray. Color coding for alignment: grey - hydrophilic, blue - positive, red - negative, orange - hydrophobic, white - non-interacting. Gaps in an alignment are denoted by a dot (.).

The online version of this article includes the following figure supplement(s) for figure 4:

**Figure supplement 1.** Per-gene interaction potentials averaged over all class I HLA molecules for TRAV genes (**A**) and TRBV genes (**B**).

**Figure supplement 2.** AIMS interaction potentials for CDR1 and CDR2 β and α chains with HLA class II molecules.

While the AIMS interaction potential only scores possible contacts between amino acid sidechains, there exists evidence in the literature that interactions with backbone atoms are often critical for overall complex stability (*Huseby et al., 2006*). However, incorporation of such information for a given interaction requires extensive knowledge of the structural details of the complex of interest, which as discussed previously, is not available for all possible TRAV/TRBV pairings. Quantification of sidechain-backbone (*Figure 5—figure supplement 2A, B*), backone-sidechain (*Figure 5—figure*

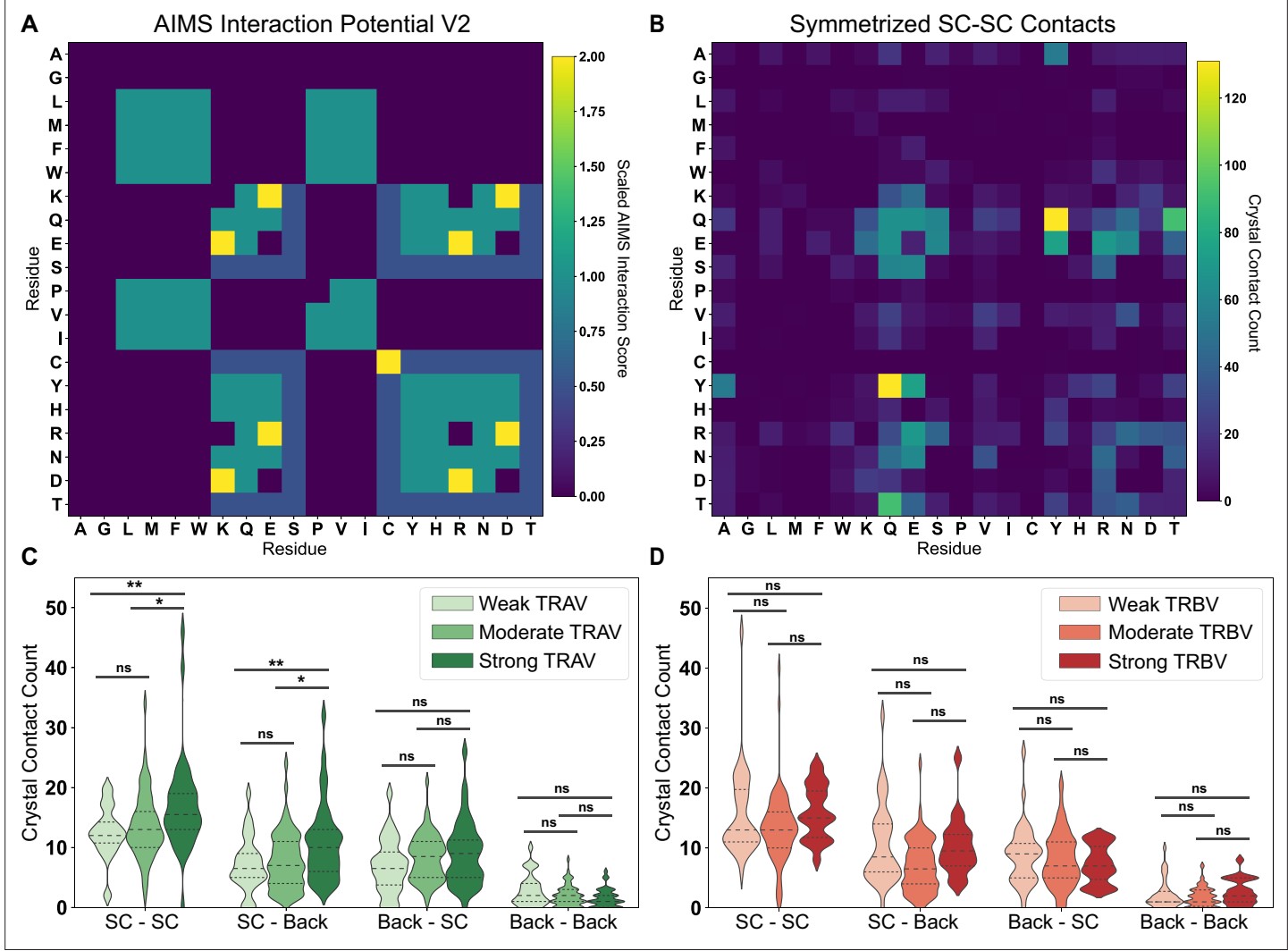

**Figure 5.** Comparison of interaction predictions to crystallized TCR-pMHC complexes. (**A**) Heat map representation of the AIMS interaction potential (version 2), normalized by mapping negative interaction potentials to 0. (**B**) A similar heat map representation for an empirical comparison to crystallized TCR-pMHC, showing a symmetrized version of the count matrix of crystal contacts between germline-encoded sidechains. Colorbars give either the AIMS interaction potential (**A**) or the raw contact count (**B**). A more rigorous quantitative comparison between the AIMS interaction potential and crystallized TCR-pMHC complexes gives these contact counts as violin plots for TRAV (**C**) or TRBV (**D**) encoded CDRs predicted to be weak (TRAV: n = 32; TRBV: n = 38), moderate (TRAV: n = 68; TRBV: n = 82), or strong (TRAV: n = 40; TRBV: n = 20) binders to MHC. Dashed inner lines give the quartiles of each distribution. Statistics determined using a non-parametric permutation test, * - $p < 0.05$, ** - $p < 0.01$, ns - not significant. X-axis Key: SC-SC:Sidechain-Sidechain, SC-Back:TCR Sidechain-MHC Backbone, Back-SC: TCR Backbone-MHC Sidechain, Back-Back:Backbone-Backbone.

The online version of this article includes the following figure supplement(s) for figure 5:

**Figure supplement 1.** Non-normalized profiles of the data in *Figure 5*.

**Figure supplement 2.** The (non-symmetrized, symmetrized) crystal contact count matrices for (**A, B**) TCR Backbone - MHC Sidechain; (**C, D**) TCR Sidechain - MHC Backbone; and (**E, F**) TCR Backbone - MHC Backbone interactions.

**Figure supplement 3.** Rendering of a TCR-MHC complex dominated by CDR3-peptide interactions (PDB:2AK4).

supplement 2C, D), and backbone-backbone (*Figure 5—figure supplement 2E, F*) contacts across TCR-pMHC complexes show some intuitive results, such as increased involvement of alanine and glycine, but other results are clearly counter-intuitive, or biased by the structures crystallized thus far. For instance, we shouldn't expect a priori that a Tyr-Gln interaction would be a common backbone-sidechain or sidechain-sidechain interaction, as both of these sidechains are relatively bulky. While the 'biases' introduced by the structures solved thus far may not be biases at all, and could in fact

be signatures of TCR recognition, it is difficult to decouple the signal from the noise without more structural information.

Despite the challenges associated with aligning model predictions with structural data, AIMS interaction scores are useful to identify those germline TRAV- and TRBV-encoded CDR loops that may make more frequent contacts with the MHC α-helices (*Figure 5C and D*). We find significant increases in the number of CDR1/2 sidechain contacts with both MHC helix sidechain and backbone atoms in TRAV sequences predicted by the AIMS interaction potential to be stronger interaction partners. While a similar agreement between interaction potential and sidechain contacts is seen for TRBV, these differences are not statistically significant. This effect is less pronounced for TCR backbone atom interactions with MHC sidechain and backbone atoms, as one would expect given that these interactions are not accounted for in the AIMS interaction potential. Interestingly, these interaction potentials show better experimental agreement and predictive power for TRAV-encoded sequences compared to TRBV-encoded sequences. This could be due either to a fundamental difference in how TCRα and TCRβ contact the MHC α-helices, or in part due to the aforementioned higher interaction potential variance for TRAV-encoded CDR loops (*Figure 4—figure supplement 1*).

However, there are still clear deviations from the predictions we can achieve at this point. In these structures, it appears that the CDR3-peptide interactions dominate the interface, overruling all other trends in germline-encoded interactions. We can consider an extreme example, looking at the germline-encoded and peptide contacts of a so-called 'super-bulged' peptide bound to MHC class I (*Figure 5—figure supplement 3*; *Tynan et al., 2005*). In this structure, the TRAV19 and TRBV6-1 encoded CDR loops, which are both identified as having a high interaction potential with HLA class I molecules, make limited germline-encoded contacts with MHC side-chains. There is only a single germline-encoded sidechain interaction between TRAV19 and the MHC α2-helix. As is evident from this structure, significant CDR3-peptide interactions are capable of strongly altering the TCR-MHC interface, regardless of which TRAV and TRBV genes are used. Absent these dominating CDR3-peptide contacts, the interaction potentials serve as a strong, structurally validated quantification of the likelihood of each germline-encoded CDR loop to interact with MHC α-helices.

## Biophysical constraints on the MHC surface create germline-encoded biases in docking angles

Having experimentally validated the interaction matrices in *Figure 4*, we explored whether there exist structural biases in the interactions between germline-encoded TCR CDR loops and MHC α-helices. First, we divided these matrices into separate datasets for each HLA group A, B, and C. We then calculated the interaction potentials across every HLA allele and TRAV or TRBV gene and identified the relative contribution of each CDR loop interacting with each HLA α-helix. From these distributions of contributions, our interaction potentials suggest that the constructive TCR β-chain interactions with the HLA α1-helix (*Figure 6A*), and perhaps to a lesser extent destructive interactions between the TCR α-chain and the MHC α1-helix (*Figure 6—figure supplement 1*), may moderately bias the binding orientation between TCR and class I HLA molecules. This reliance on TCR β-chain interactions to guide the overall TCR-MHC interaction has been suggested previously in the literature based upon multiple early observations in mice (*Marrack and Kappler, 1987*). It is also in line with previous findings that CDR1 and CDR2 of the TCR β-chain are key drivers of the TCR-MHC docking orientation (*Feng et al., 2007*; *Garcia et al., 2009*). Conversely, our calculations suggest the α2-helix is only weakly involved in the determination of binding orientation (*Figure 6B*). The exposed residues on the α2-helix of HLA class I molecules are enriched in alanine and glycine relative to the β1-helix, which are highly unlikely to be involved in a specific, orientation-altering productive interaction.

Interestingly, the TCR-HLA interaction potential appears flipped for class II molecules: interactions between TCR germline-encoded regions and HLA class II molecules are broadly driven instead by the β-chain helix (*Figure 6C and D*). We note that the interaction potential incorrectly predicts that the TCR β-chain CDR loops should be the strongest interaction partners with the HLA class II β-chain helix. Nearly every crystallized TCR-HLA class II complex solved thus far adopts the canonical docking orientation whereby the TCR β-chain binds to the HLA α-chain helix, while the TCR α-chain binds to the HLA β-chain helix, with few notable exceptions (*Beringer et al., 2015*). It is important to note that these interaction potentials take an unbiased approach, calculating every possible interaction between TCR and MHC residues to produce this final score. Therefore, this inconsistency between our

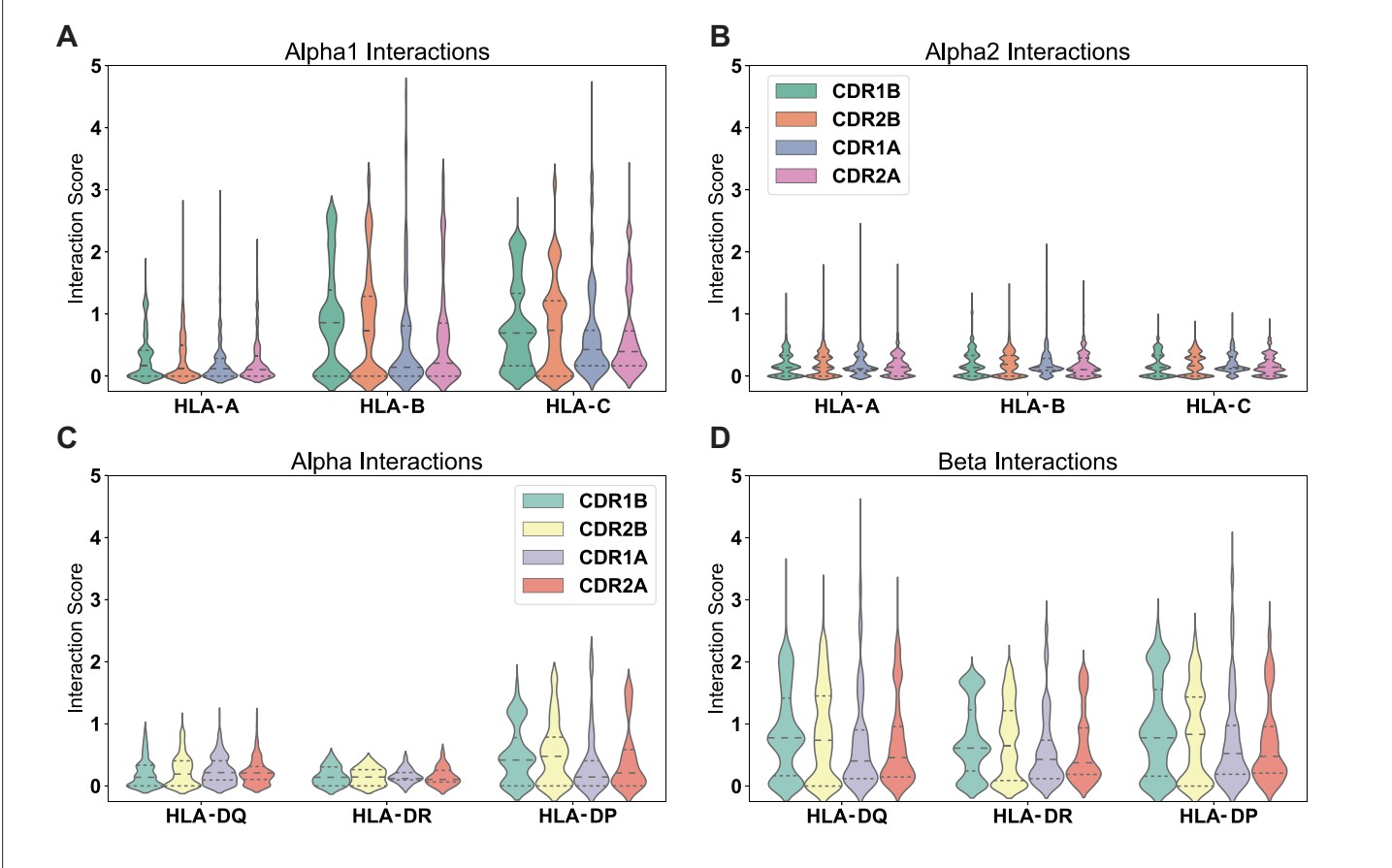

**Figure 6.** Positive interactions between each germline-encoded TCR CDR loop and all HLA alleles. Violin plots give distribution of interaction scores, with dashed lines separating quartiles of the distributions. The individual interaction potentials are shown for the individual CDR loop interactions with MHC Class I α1-helix (**A**), MHC Class I α2-helix (**B**), MHC Class II α-helix (**C**), and MHC Class II β-helix (**D**). Reported values are averages for each individual TRAV or TRBV gene and their interactions with each individual MHC allele over both CDR loops. Number of points in each violin plot given as follows (MHC: n = #TRAV, #TRBV): HLA-A: n = 10755, 11472; HLA-B: n = 17010, 18144; HLA-C: n = 11925, 12720; HLA-DQα: n=945, 1008; HLA-DRα: n = 45, 48; HLA-DPα: n = 990, 1056; HLA-DQβ: n = 5985, 6384; HLA-DRβ: n = 8550, 9120; HLA-DPβ: n=4860, 5184.

The online version of this article includes the following figure supplement(s) for figure 6:

**Figure supplement 1.** AIMS clash potentials calculated for all possible CDR-MHC helix interactions.

interaction potential distributions and crystallized structures may arise from variations across a given MHC helix obscuring the interaction potentials within the canonical binding region, highlighting the need for a more precise approach. To more carefully characterize the CDR loop interactions with MHC α-helices and attempt to alleviate this inconsistency, we isolated the contributions of each individual residue in the HLA class I and class II sequence alignments to the interaction potentials (*Figure 7*).

*Figure 7A and B* show a structural visualization of the amino acids of interest on the class I (*Figure 7A*) and class II (*Figure 7B*) HLA molecules. These TCR-exposed residues are colored based approximately upon the register a TCR would adopt upon binding each subset, whereby TCRs adopting the canonical binding orientation would contact the residues colored in cyan and pink, and non-canonical binders would contact the blue and red residues. For both HLA class I (*Figure 7C*, top) and class II (*Figure 7D*, top), we found a conserved pattern of specific regions of peak interaction potentials surrounded by immediate drop-offs in these interaction potentials. In all but the HLA class I α1-helix, the interaction potentials adopt a distinctly bimodal distribution over the surface of the HLA molecules. In the case of class I molecules, our interaction analysis suggests that the TCR β-chain binds to the HLA class I α1-helix nearly indiscriminately, with potential to form many interactions. The TCR α-chain then contributes to the orientation of the interactions by binding to the HLA class I α2-helix in one of two discrete regions. Conversely, interactions between the αβ TCR and HLA class II molecules

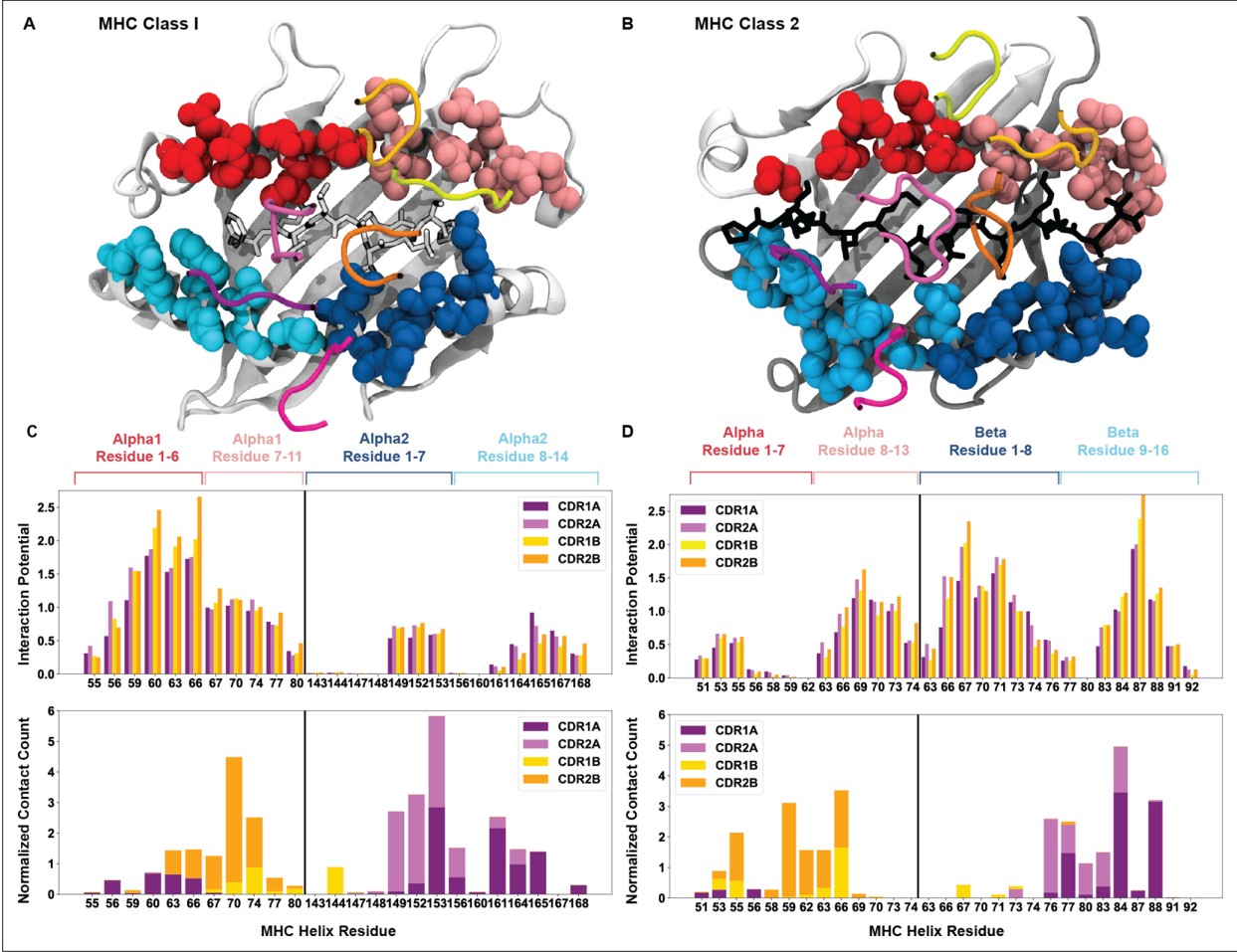

**Figure 7.** Breakdown of the interaction potential for every exposed residue on HLA surfaces. Color coded structures of HLA class I (**A**, PDB: 6MTM) and class II (**B**, PDB: 1J8H) provide the structural context to the bioinformatic results. The colors of the amino acid side-chains match the labels of the later panels. The averaged interaction potential across TRAV and TRBV genes with each individual MHC helix is shown in the top panels for HLA class I (**C**) and HLA class II (**D**). Lower panels of (**C**) and (**D**) compare these interaction potentials to total crystal contacts with these same residues (n = 149 HLA class I structures, n = 44 HLA class II structures).

appear to have a more defined register for binding, with both the HLA class II α-chain and β-chain interactions with TCR forming distinct bimodal distributions.

We then compared these by-residue interaction breakdowns to crystallized TCR-pMHC structures, counting the number of side-chain interactions between CDR loops and the solvent-exposed residues of the MHC helices. For HLA class I (*Figure 7C*, bottom), we found good agreement between our predictions and the crystal contacts. Particularly for contacts with the α2-helix, we see that the majority of CDR1/2α contacts fall directly within the predicted regions of non-zero interaction potential. Comparisons to HLA class II (*Figure 7D*, bottom) are more difficult, again due to the relative dearth of crystallized structures. Overall, these results further contextualize the seemingly inconsistent findings of *Figure 6D*. While the interactions with the class II β-chain are stronger, we see that the average over all interactions with the α-chain are subdued by the very weak interaction potential of the non-canonical binding region. Within the canonical binding regions of the class II molecule, the TCR α-chain and β-chain CDR loops have similar binding potentials with either α-helix.

## The binding register of MHC molecules is well conserved by areas of low interaction potential

The identification of these localized regions of low interaction potential contrasts with what has been postulated previously to be the root of the conserved TCR-pMHC docking orientation. Instead of

confirming that conserved interactions determine the docking orientation, our results suggested that regions that are less likely to contact the TCR generate this preference. To further characterize these regions of low interaction potential flanked by regions of increased interaction potential on the MHC surface, we examined position-sensitive biophysical properties to characterize the exposed residues

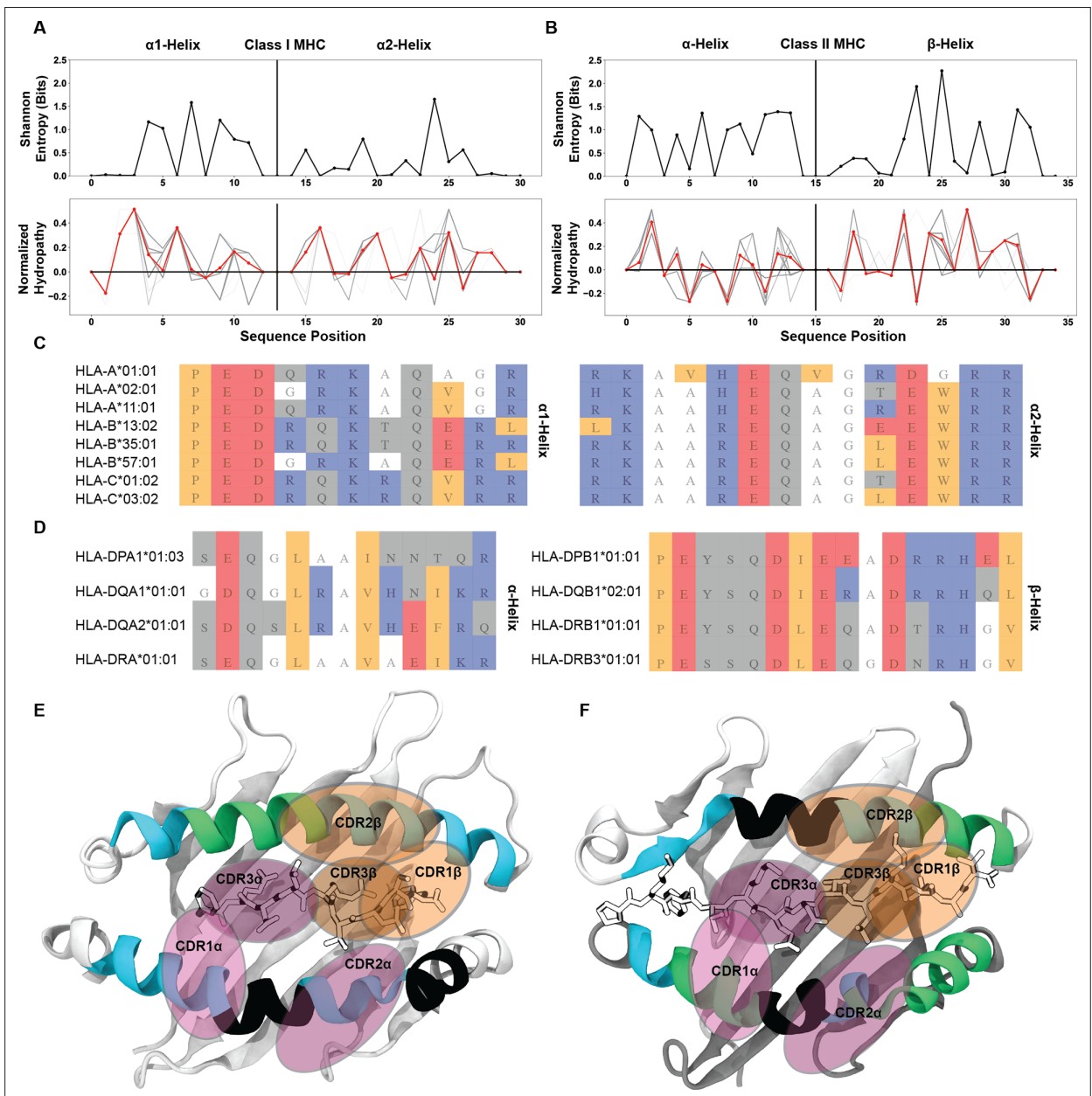

**Figure 8.** Identification of well conserved regions of low interaction potential finalize a working model for the root cause of canonical TCR-MHC docking orientations. Position-sensitive Shannon entropy (top) and normalized amino acid hydropathy (bottom) for class I (**A**) and class II (**B**) HLA molecules. Red lines in the hydropathy plots indicate an average over all HLA molecules, while gray lines give the position-sensitive biophysical properties of individual molecules. Alignments of class I (**C**) or class II (**D**) HLA alleles from a subsampling of parental alleles, colored by biophysical property. Color coding for alignment: grey - hydrophilic, blue - positively charged, red - negatively charged, orange - hydrophobic, white - non-interacting. Renders of class I (**E**, PDB: 6MTM) and class II (**F**, PDB: 1J8H) HLA molecules with α-helices colored by interaction potential. Green - regions of high interaction potential, cyan - regions of moderate interaction potential, black - regions of negligible interaction potential. Orange ovals give probable contact regions for TCRβ, while purple ovals give probable contact regions for TCRα, defining canonical docking orientations.

The online version of this article includes the following figure supplement(s) for figure 8:

**Figure supplement 1.** Across a range of mammalian species, the regions of low interaction potential on class I and class II MHC α-helices are very well conserved.

of the MHC α-helices. Specifically, we directly compared the position-sensitive Shannon entropy to the position-sensitive hydropathy of these residues across all MHC class I (*Figure 8A*) and class II (*Figure 8B*) molecules.

We found that in many of the regions where the hydropathy is at or near zero, meaning the region is neither hydrophilic nor hydrophobic, there is drop in the entropy to a value near zero. These residues with a hydropathy near zero likewise have an interaction potential near zero. This suggests that these residues with a lower interaction potential are well-conserved across HLA molecules. Indeed, looking again at alignments of HLA class I (*Figure 8C*) and class II (*Figure 8D*) sequences, we found that these regions of alanine and glycine usage are well conserved in a sampling of the so-called 'parental' HLA alleles (*Robinson et al., 2017*). Further inspection of a matrix encoding of a broader set of mammalian MHC molecules (*Figure 8—figure supplement 1*) showed that alanine and glycine are well conserved in specific regions across species in the α2-helix of MHC class I molecules, as well as in both helices of MHC class II molecules.

These observations completed our final model for the source of bias in the conserved TCR-MHC docking orientation (*Figure 8E and F*). The structures of HLA class I (*Figure 8E*) and HLA class II (*Figure 8F*) are colored based on their average interaction potentials as shown in *Figure 7*. We readily see the well-conserved areas of negligible interaction potential (black) occur near the centers of the α-helices for each MHC. By overlaying typical CDR contact regions, we can visualize how these regions of low interaction potential may be capable of dictating the docking angle. The α2 and β-chain helices, due in part to the centrally located region of reduced interaction potential and in part due to the kink in helix, appear to play a key role in determining the conserved docking angle.

## Discussion

The concept of evolutionarily conserved interactions guiding immune recognition dates back half a century to work by *Jerne, 1971*. Despite predating the first TCR-MHC structures the prediction has withstood the test of time. Canonical TCR-MHC docking orientations have repeatedly been observed and specific instances of evolutionarily conserved contacts have been identified in a subset of structures (*Feng et al., 2007*; *Blevins et al., 2016*; *Scott-Browne et al., 2011*). However, the identification of TCRs that are capable of binding to non-MHC ligands (*Van Laethem et al., 2007*) or in a reversed docking orientation (*Beringer et al., 2015*; *Gras et al., 2016*; *Zareie et al., 2021*) call into question the general validity of these findings of evolutionary conservation and highlight the need for systematic analyses covering the entire space of TRAV, TRBV, and MHC alleles. Even without these eccentric exceptions, evidence for evolutionarily conserved interactions have only been convincingly shown for a small subset of TRAV and TRBV genes. Here, we report the results of the first systematic study across these variable alleles.

### Diversity analysis rules out the existence of specific conserved contacts for a majority of TRAV- and TRBV-encoded CDR loops

Using AIMS, a recently developed software (*Boughter et al., 2020*; *Boughter and Meier-Schellersheim, 2023*) for encoding and analyzing amino acid sequences and their biophysical properties in the context of molecular interactions, we found that across all tested TRAV, TRBV, and HLA alleles, there is no evidence of strongly conserved germline-encoded interactions that exist for all possible combinations of TRAV/TRBV genes and HLA polymorphisms. The sequence-level diversity in the germline-encoded TCR CDR loops alone makes such interactions highly unlikely. Biophysical analysis further confirmed that the wide range of physical properties of these highly diverse germline-encoded CDR loops can lead to steric conflicts in regions where other germline genes create productive interaction networks. These findings suggest that conserved TCR-MHC interactions spanning species identified thus far (*Blevins et al., 2016*; *Scott-Browne et al., 2009*; *Scott-Browne et al., 2011*; *Dai et al., 2008*) may be rare across the entirety of the immune repertoire. To characterize these TCR-MHC germline-encoded interactions outside the lens of this evolutionary context, we employed a biophysical scoring function to quantify the pairwise interaction potential for every TRAV/HLA and TRBV/HLA pair.

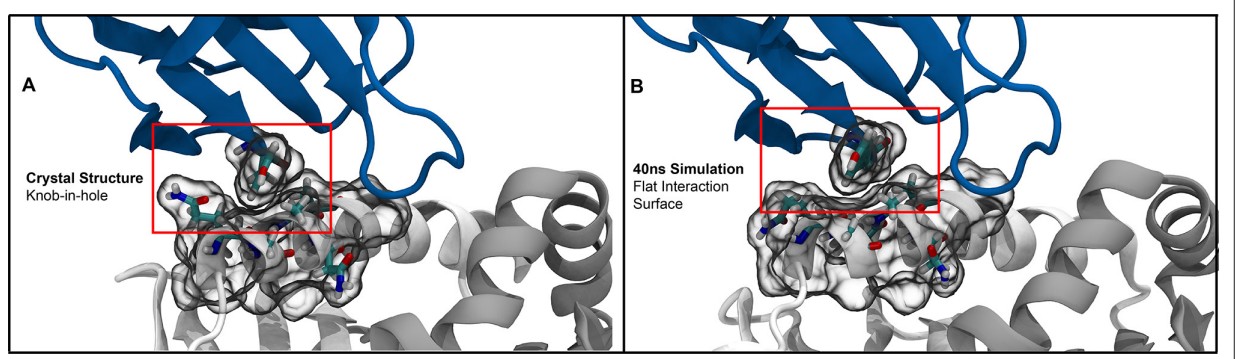

**Figure 9.** Molecular simulations of the so-called "knob-in-hole" interaction (*Garcia, 2012*) highlight the dynamic nature of protein sidechains. (**A**) The starting crystal structure (PDB:1FYT) suggests a tight packing between TYR50 on CDR2β and GLN57 and ALA61 on the class II α-chain. (**B**) Short all-atom simulations show that this suggestive tight packing is due to the static nature of crystal structures, with GLN57 freely adopting alternate conformations over the course of the simulation.

The online version of this article includes the following figure supplement(s) for figure 9:

**Figure supplement 1.** Triplicate molecular dynamics simulations highlight the short-lived nature of the "knob-in-hole" interaction.

## Biophysically compatible regions of MHC play a key role in determining the canonical docking orientation

These by-allele interaction potentials can further be broken down on a per-residue basis for each MHC α-helix. Surprisingly, this analysis may suggest why, in spite of the observed sequence variability, we find rather well-conserved TCR-pMHC docking orientations. Across nearly all tested human polymorphisms, we found that class I and class II MHC helices are composed of moderately diverse regions of increased interaction potential interrupted by centrally located, well-conserved regions of low interaction potential. These regions of low interaction potential define a conserved docking orientation, with room for slight variability in orientation about the flanking regions of higher interaction potential. Our findings do not contradict the previously proposed concept of 'interaction codons' whereby specific TCR-MHC pairs are predisposed to interact in multiple registers dependent upon the peptide bound to the MHC (*Adams et al., 2016*; *Feng et al., 2007*; *Garcia et al., 2009*). However, these codons as currently posited suggest the existence of multiple rigid docking modes, consistent with their ideation deriving from solved crystal structures. The results of our analyses suggest a broader, dynamic interpretation of the TCR-pMHC interface (*Baker et al., 2012*; *Devlin et al., 2020*; *Smith et al., 2021*; *Borbulevych et al., 2009*; *Scott et al., 2011*), with the TCR CDR loops sampling local regions of increased interaction potential on the MHC surface, utilizing a more opportunistic approach to binding. Further, while the interaction codon hypothesis suggests co-evolved interaction interfaces between the TCR and MHC, our analysis suggests that each TRAV-TRBV pairing finds a unique approach to binding within the constraints permitted by the MHC molecular surface. In other words, the MHC molecule largely defines the interface.

In further extending these dynamic interpretations of the TCR-pMHC interface, all-atom molecular dynamics simulations highlight the inherently fleeting nature of sidechain interactions. As a case study, we can consider a well-studied tyrosine-alanine-glutamine interaction which appears to be an example of exceptional shape complementarity between the TCR and MHC surfaces (*Figure 9A*). Triplicate, all-atom molecular dynamics simulations of a TCR-pMHC complex (PDB: 1FYT *Hennecke et al., 2000*) highlight the short lifetimes of such intricate molecular interaction networks (*Figure 9B*). Over the course of these simulations (*Figure 9—figure supplement 1*), the intrinsic motility of the glutamine sidechain in the interface significantly alters the interpretation of the tyrosine-alanine interaction from a well-evolved notch reserved for TCR binding to merely a consequence of other factors determined strongly by the potentially more dominant CDR3 interactions with the interface.

These simulations are hardly the first investigations into TCR and MHC dynamics (*Baker et al., 2012*), but highlight that care should be taken when discussing sidechain interactions within these protein-protein complexes. While we focus briefly on only a trio of interacting residues, one should generally expect that sidechain bonds in crystal structures are unlikely to persist over the entire

microsecond-scale binding process. Further, crystal structures should be considered snapshots in a local energy minimum, which at near-physiological temperatures likely sample a wide range of contacts (*Mei et al., 2020*; *Bradford et al., 2021*). These dynamic interpretations are consistent with the broad biophysical compatibility suggested by the interaction potential results.

## Quantification of germline-encoded CDR interactions with MHC α-helices identifies V-gene-dependent recognition strategies

The granular details of the TCR-MHC interaction potential provides an antigen-independent prediction of the relative contributions of the germline-encoded CDR loops to the overall interaction interface. The CDR loops encoded by these TRAV and TRBV genes each have largely distinct binding strategies with MHC. While many V-genes encode CDR1 and CDR2 loops with equal interaction potentials with MHC α-helices, a small subset appears to favor one or the other for binding to MHC molecules. Importantly, these preferences seem to be accompanied by compensatory contributions to binding; the interaction potentials of TRBV6-1, 6–2, and 6–3 have negligible CDR2β contributions to binding, but in turn have a stronger propensity for MHC binding via CDR1β (*Figure 4*.). These compensatory contributions to binding are suggestive of a mix-and-match strategy for peptide-MHC complex recognition, consistent with previous findings highlighting the strong dependence of complex geometry on CDR3 residues (*Stadinski et al., 2014*; *Piepenbrink et al., 2013*; *Lu et al., 2019*).

What these previous studies and our computational results suggest is that for a T cell to recognize a given antigen, the CDR3 loops must bind sufficiently strongly to the peptide, while the CDR1 and CDR2 loops can provide a basal level of support for the more complex-specific CDR3-induced binding. If all V-genes encoded CDR1 and CDR2 that dominate the interaction with strong, specific contacts with MHC, they may well impede the exquisite specificity of the overall TCR-pMHC interaction determined by the CDR3 loops. The diversity in the V-genes instead permits a range of recognition strategies independent of, and complementary to, the V(D)J recombination-generated CDR3 loops.

## Biophysical compatibility between germline encoded TCR sequences and MHC does not rule out alternative binding modes

The general TCR-MHC compatibility as suggested by our results does not rule out the possibility of reversed-docking TCRs, suggesting both evolution and selection play a strong role in the conservation of the canonical docking orientation. While it is unclear how prevalent these reversed-docking TCRs are in the pre-selected immune repertoire, it is clear that they are deficient in their ability to signal due to the distance-sensitivity of the coreceptor-associated Lck in interactions with the CD3 signaling complex (*Adams et al., 2011*; *Gras et al., 2016*; *Zareie et al., 2021*). In our analysis of the biophysical compatibility between germline-encoded TCR sequences and their potential interaction sites on the MHC molecule, we found strong indications that this may be a necessary coevolutionary tradeoff. Over evolutionary history, TCR and MHC molecules have had to balance a conservation of canonical modes of contact with the need to maintain sequence variability in the face of evolutionary pressure. This pressure, exerted by pathogenic challenges, may have forced a drift away from rigid coevolution at the level of amino acid sequences, necessitating this loose compatibility, even at the cost of a major subset of receptors being destined for thymic deletion due to an inability to generate productive signal.

New results further suggest the extent of the germline interaction permissiveness, with a class-mismatched CD4+ T cell capable of binding to and being activated by MHC class I, albeit with a slightly abnormal, but not reversed, docking orientation (*Singh et al., 2022*). These results further highlight the opportunism, expanding on previous work in this space (*Blevins et al., 2016*), of TCR interactions in general, where the 'rules' of TCR-pMHC binding seem to be more like guidelines. The literature has long focused on rules of interaction and commonalities between structures (*Ysern et al., 1998*; *Al-Lazikani et al., 2000*), which have been very helpful in guiding research over the past few decades. However, results such as the class-mismatched TCR, reversed docking TCRs, and super-bulged peptide suggest that perhaps such TCR-MHC specific rules may be too restrictive, and that these interactions may frequently involve more opportunistic configurations that call for unbiased evaluation.

## Towards generating a generalizable model for TCR-pMHC recognition

The ideal approach for studying TCR-pMHC interactions would involve generating a comprehensive set of crystallized structures, exhaustive biochemical experiments to pinpoint contributions to binding affinity and kinetics, and activation assays to thoroughly understand the nuances that underlie complex formation. While such thorough efforts have been undertaken to understand structural strategies of binding to HLA-A2 (*Blevins et al., 2016*), the potential evolutionary conservation of TRBV8-2 in binding MHC molecules (*Scott-Browne et al., 2009*; *Scott-Browne et al., 2011*; *Garcia, 2012*; *Feng et al., 2007*), and the by-residue contributions of binding of the A6 TCR (*Piepenbrink et al., 2013*), the diversity inherent to the TCR-pMHC interaction makes such efforts impossible to scale across all possible binding partners.

The work presented here cannot replace these comprehensive experimental techniques. However, it can provide a good first estimate of how well given TCR-MHC pairs may bind. Absent rich experimental data on major parts of the TCR and MHC repertoire space, can we attempt to approximate how a fictitious TCR-MHC interaction would form independent of peptide and CDR3? Comparisons of our computational results to experiment suggest that yes, in fact, we can generate some strong approximations to build off of, with clear deviations from these predictions largely driven by outlier structures. While the simplified interaction potential we have used for the analyses presented here performed surprisingly well in its predictions for overall protein-protein binding propensities, it is clear that it leaves room for improvement, in particular with regard to aspects such as the relative spatial positioning of the interacting structures. As we continue to build these interaction potentials and the AIMS software as a whole, we hope to continually add modular improvements to consider how these initial germline interaction assumptions are altered by peptide and CDR3 to build more predictive tools for interactions and binding.

## Ideas and speculation: multi-modal recognition strategies provide a mechanism for autoimmune and non-canonical TCRs

The existence of V-genes with weakly interacting CDR loops that have little to no propensity for binding to the MHC α-helices suggests surprising modes of immune recognition. Among them, those with the weakest interactions may, in fact, be the most interesting. V-genes including TRBV7-2, 7–3, 20–1, and 29–1 have no potential for interaction with MHC, and are associated with celiac disease (*Gunnarsen et al., 2017*; *Qiao et al., 2014*) and multiple sclerosis (*Sethi et al., 2013*). Previous studies have shown that TCR-pMHC interactions dominated by non-germline-encoded CDR3 contacts enable TCR cross-reactivity, and in some cases autoimmunity (*Ciacchi et al., 2022*; *Petersen et al., 2014*; *Petersen et al., 2020*; *Sethi et al., 2013*; *Hahn et al., 2005*). This, coupled with the aforementioned germline associations with autoimmune disorders, leads to the intriguing possibility that interactions between germline-encoded TCR regions and the MHC provide a framework for reliable peptide differentiation whereas the lack of such interactions, while granting more flexibility, may lead to higher instances of misguided T cell activation.

Thinking of the entire TCR, not just the germline-encoded CDR loops, as a necessarily cross-reactive (*Mason, 1998*; *Sewell, 2012*) opportunistic binder (*Singh et al., 2022*) provides a potential mechanism for the thymic escape of autoimmune TCRs and reversed docking TCRs. Consider a TCR utilizing TRAV1-1 and TRBV7-3, a V-gene pairing with the lowest possible MHC interaction potential. While a few CDR1 and CDR2 backbone interactions may contribute to the overall complex stability, the CDR3 loops would be largely responsible for forming strong interactions with MHC. In the absence of the germline-encoded binding framework, such strong dependence on CDR3-mediated interactions would make antigen recognition similarly dependent on the CDR3 loop conformation. If this TCR were to encounter a potentially strongly binding (auto-)antigen in the thymus, its probability of being negatively selected would somewhat stochastically depend on the conformations adopted by those TCRs when binding to the pMHC complexes, thereby increasing the chances for thymic escape and subsequent autoimmune events in the periphery.

While over-reliance on CDR3 may bias TCRs towards auto- and cross-reactivity, the role of the MHC molecule in autoimmunity remains unclear from these results. In contexts such as celiac disease, a particular allele (HLA-DQ2.5) is enriched in patients with the disease (*Qiao et al., 2014*; *Gunnarsen et al., 2017*). Given that the interaction potentials largely predict similar interaction strengths across HLA molecules (*Figure 4*, *Figure 4—figure supplement 2*) such enrichment cannot yet be explained

by AIMS. However, considering that the majority of diversity across HLA alleles is concentrated in the peptide binding regions (*Figure 2*), these correlations between HLA molecules and disease may largely be related to the peptides presented by these molecules, as has been suggested previously (*Ishigaki et al., 2022*). In trying to understand how allelic variation alters peptide presentation, and how this in turn impacts the onset of autoimmunity, the lack of strong rules for binding again complicates the problem. While HLA molecules have amino acid preferences at certain anchor positions, there are many exceptions to these 'rules' (*Nguyen et al., 2021*), and mutations to peptides that should improve stability in the HLA binding pocket can have unintended consequences on T cell activation (*Smith et al., 2021*). The substantial diversity of possible presented peptides makes the systematic analysis of this aspect of the autoimmunity problem exceptionally challenging.

## Methods

The AIMS software package used to generate the analysis in this manuscript is available here, including the original Jupyter Notebook used to generate the Figures in this manuscript as well as a generalized Notebook and a python-based GUI application for analysis of novel datasets. The AIMS software is constantly evolving, so to ensure exact recapitulation of results presented here AIMS v0.8 should be used. Detailed descriptions of the analysis and the instructions for use can be found at https://aims-doc.readthedocs.io.

### Repertoire analysis using AIMS

As with all AIMS analyses, the first step is to encode each sequence into an AIMS-compatible matrix. In this encoding each amino acid in structurally conserved regions is represented as a number 1–21, with zeros padding gaps between these structurally conserved features. The encoding is straightforward for the TCR sequences, with only the germline encoded regions of the CDR loops included for each gene. For the MHC encoding, only the structurally relevant amino acids are included for optimal alignment of each unique sequence. In this case, the structurally relevant amino acids of the class I (or class II) molecules were divided into three distinct groups; the TCR-exposed residues of the α1 (or α) helix, the TCR-exposed residues of the α2 (or β) helix, and the peptide-contacting residues of the given MHC.

Standard AIMS analysis, including calculation of information-theoretic metrics and position-sensitive biophysical properties, is then derived from these AIMS-encoded matrices. The AIMS interaction scoring is based on a matrix that quantifies a basic pairwise interaction scheme (*Supplementary file 1*), whereby productive amino acid interactions (salt bridges, hydrogen bonds) are scored positively while destructive interactions (hydrophilic-hydrophobic and like-charge clashes) are scored negatively. The first version of this interaction scoring matrix has previously been used to classify interacting and non-interacting molecular partners with a distinguishability of nearly 80% (*Nandigrami et al., 2022*).

In calculating the interaction score, we assume that productive contacts are only made by the side-chains of the interacting residues. This simplification does not capture all TCR-pMHC complex contacts, but here we are looking for selectivity enforced by specific TCR-MHC interactions mediated by side-chains in a structure-free manner. Further, given that any single pair of amino acids on adjacent interfaces of protein binding partners can potentially form strong interactions without being meaningful for the formation of a given complex, we require that any productive interaction include a trigram of at least weakly interacting residues.

For example, for the TCR sequence YNNKEL, we break the sequence into trigrams YNN, NNK, NKE, KEL. A portion of an MHC α1 helix with the sequence EDQRKA is similarly broken intro trigrams EDQ, DQR, QRK, RKA. While AIMS takes into account every possible interaction combination of these triads, the interaction partners TCR:NNK and MHC:EDQ would be scored positively, while TCR:KEL and MHC:RKA would be discounted as an unfavorable interaction. In the former case the scores according to the AIMS interaction matrix come out to [+1,+1,+1] while in the latter case the score [−2,+2,+0] includes clashing residues, and is therefore not counted as a possible interacting triad. This scoring methodology defines our search parameters for broad biophysical compatibility between TCR CDR loops and the MHC helices, from which we can generate averaged interaction potentials for every TRAV and TRBV gene with every possible HLA allele.

## Sequence processing

All sequences used in this work are derived from the ImMunoGeneTics (IMGT) (*Brochet et al., 2008*; *Lefranc, 2014*) database of TRAV and TRBV alleles (https://www.imgt.org/IMGTrepertoire/Proteins) from the following organisms: *Bos taurus* (cow), *Capra hircus* (goat), *Aotus nancymaae* (Nancy Ma's Night Monkey), *Mus musculus* (mouse), *Macaca mulatta* (rhesus macaque), and *Ovis aries* (sheep). MHC sequences are derived from the IMGT-HLA database containing all identified HLA alleles from human and a wide range of organisms (https://www.ebi.ac.uk/ipd/imgt/hla/download) (*Mack et al., 2013*). Only unique, productive sequences are included, which excludes all open reading frames and pseudogenes. Specific structural features are identified from these productive sequences for each molecular species. For TCRs, these structural features are the complementarity determining region (CDR) loops as defined by IMGT. For MHC molecules, these are the TCR-exposed residues of the MHC α-helices as defined in *Bjorkman and Parham, 1990* for class I and as identified in visual molecular dynamics (VMD) (*Humphrey et al., 1996*) for class II. Class I identifications were likewise validated in VMD. For comparison to non-TCR contacting residues, the peptide-contacting residues were also included for class I molecules, again from identification in *Bjorkman and Parham, 1990*. All downstream sequence processing was done from these regions of each TCR and MHC sequence.

## Structural comparison processing pipeline

We accessed 149 human TCR-pMHC class I structures and 44 human TCR-pMHC class II structures with PDB IDs drawn from the TCR3D database (*Gowthaman and Pierce, 2019*) and each PDB loaded into python using the mdtraj package (*McGibbon et al., 2015*). After parsing the TCR3D database for degenerate structures and improperly deposited PDBs, we compiled a final dataset of 96 class I structures and 36 class II structures. We extracted CDR-MHC distances for all contacts. The interaction cutoff is set to 4.5Å for van der Waals interactions and 6Å for charged interactions. Oxygen-Oxygen/Nitrogen-Nitrogen were only counted as productive electrostatic bonding pairs if Ser/Thr/Tyr were involved in O-O contacts or if His was involved in N-N contacts. Through a random selection of 10 complex structures analyzed using this pipeline, we were able to validate structural distance measurements using VMD and found that 100% of the identified contacts match those in structures. All code for sequence processing and structural analysis is included in a separate repository, PRESTO (PaRsEr of Solved Tcr cOmplexes). Code for reproducing the analysis can be found here.

## Information theoretic calculations

Information theory, a theory classically applied to communication across noisy channels, is incredibly versatile in its applications and has been applied with success to a range of immunological problems (*Shannon, 1948*; *Román-Roldán et al., 1996*; *Cheong et al., 2011*; *Vinga, 2014*; *Mora et al., 2010*; *Murugan et al., 2012*). In this work, we utilize two powerful concepts from information theory, namely Shannon entropy and mutual information. Shannon entropy, in its simplest form, can be used as a proxy for the diversity in a given input population. This entropy, denoted as H, has the general form:

$$H(X) = - \sum_{X} p(x) \log_2 p(x)$$

(1)

where $p(x)$ is the occurrence probability of a given event, and $X$ is the set of all events. We can then calculate this entropy at every position along the CDR loops or MHC α-helices, where $X$ is the set of all amino acids, and $p(x)$ is the probability of seeing a specific amino acid at the given position. In other words we want to determine, for a given site in a CDR loop or MHC helix, how much diversity (or entropy) is present. Given there are only 20 amino acids used in naturally derived sequences, we can calculate a theoretical maximum entropy of 4.32 bits, which assumes that every amino acid occurs at a given position with equal probability.

Importantly, from this entropy we can calculate an equally interesting property of the dataset, namely the mutual information. Mutual information is similar, but not identical to, correlation. Whereas correlations are required to be linear, if two amino acids vary in any linked way, this will be reflected as an increase in mutual information. In this work, mutual information $I(X;Y)$ is calculated by subtracting the Shannon entropy described above from the conditional Shannon entropy $H(X|Y)$ at each given position as seen in *Equations 2; 3*:

$$H(X|Y) = -\sum_{y \in Y} p(y) \sum_{x \in X} p(x|y) \log_2 p(x|y) \tag{2}$$

$$I(X;Y) = H(X) - H(X|Y) \tag{3}$$

These equations for the entropy and the mutual information are used for all information theoretic calculations in this manuscript, but special consideration of the TCR and MHC sequences must be applied in order to reformulate these sequences into the classic input/output framework necessary for information theory. Keeping with the terminology of information theory, we would define the TCR-pMHC interactions as a 'communication channel', and thus think of each TCR sequence as a given input. In this picture, each HLA allele would be seen as a corresponding output. If there exists some systematic relationship between the amino acids in the TCR and HLA sequences, we would see a significant influence of TCR sequence variations on HLA sequences, and this should manifest as an increase in the mutual information.

To calculate this mutual information, we start with the assumption that if the concept of evolutionary conservation of TCR-pMHC interactions is correct, then one should assume that every TCR should interact with every HLA allele. Humans largely possess the same TRAV and TRBV alleles, but each individual possesses a maximum of 12 HLA alleles. We expect that specific alleles that are unable to enforce the supposed evolutionary rules for canonical docking will not be allowed to persist in the population. Continuing from this assumption we then subsample the data and calculate the mutual information on this subsampled dataset. Each TRAV and TRBV allele (the input) is matched with a single HLA allele (the output), and the mutual information is calculated for these pairings. This process is repeated 1000 times and the average mutual information is reported. Further validation of our approach on non-human organisms was carefully formulated to only reclassify TRAV/TRBV alleles and MHC sequences as input/output sequences from the same organismal source. The mutual information calculation was then carried out across organisms, with this within-species architecture conserved.

## All-atom MD simulations

All simulations performed were prepared using the CHARMM-GUI Input Generator (*Jo et al., 2007*; *Jo et al., 2008*; *Lee et al., 2016*). Three replicas of PDB 1FYT were fully hydrated with TIP3P water molecules and neutralized with 0.15 M KCl. All simulations were carried out in simulation boxes with periodic boundary conditions using the additive PARAM36m force field from the CHARMM (Chemistry at HARvard Macromolecular Mechanics; *Brooks et al., 2009*). Simulations were run on GPUs using the AMBER architecture for a simulated time of 80 ns, with a 4 fs time step and hydrogen mass repartitioning at 303.15 K (*Hopkins et al., 2015*). For all simulated systems run on the Locus Computing Cluster at the NIAID, NIH, at least two replicas were run to confirm the results' independence on initial velocity assignments. Data were analyzed using a customized python package.

To reduce simulation size while maintaining the molecular architecture enforced upon TCR-pMHC complex formation, we removed the TCR Cα and Cβ regions of the TCR, as well as the α2 and β2 regions of the class II MHC molecule. To replace the lost stability introduced by these regions, weak restraints (0.1 kcal/[mol*Å$^2$]) were applied to the β-strands of the MHC class II platform domain and the C-termini of the Vα and Vβ domains of the TCR. Restraints were only applied to the carbon-α atoms of these regions.

## Acknowledgements

This work was supported by the intramural program of the National Institute of Allergy and Infectious Diseases (NIAID), NIH. We would like to thank David Margulies, Pamela Schwartzberg, Alexander Brown, and Charles Dulberger for their insightful comments.

## Additional information

### Funding

| Funder | Grant reference number | Author |
|---|---|---|
| National Institutes of Health | ZIA AI001076-16 | Martin Meier-Schellersheim |

The funders had no role in study design, data collection and interpretation, or the decision to submit the work for publication.

### Author contributions

Christopher T Boughter, Conceptualization, Resources, Data curation, Software, Formal analysis, Validation, Investigation, Visualization, Methodology, Writing - original draft, Writing – review and editing; Martin Meier-Schellersheim, Supervision, Funding acquisition, Project administration, Writing – review and editing, Conceptualization, Resources

### Author ORCIDs

Christopher T Boughter (i) http://orcid.org/0000-0002-7106-4699
Martin Meier-Schellersheim (i) http://orcid.org/0000-0002-8754-6377

### Decision letter and Author response

Decision letter https://doi.org/10.7554/eLife.90681.sa1
Author response https://doi.org/10.7554/eLife.90681.sa2

## Additional files

### Supplementary files

• Supplementary file 1. Table used for the second version of the AIMS scoring of pairwise amino acid interactions. The table attempts to recapitulate the interactions between amino acids at the level of an introductory biochemistry course.

• Supplementary file 2. Key for *Figure 4* and *Figure 4—figure supplement 1* relating the numbers on the X-axis of each plot to the corresponding TRAV or TRBV gene. Note, the pairing of TRAV and TRBV genes to a specific X-axis number has no meaningful relation. Genes are listed in the same order as found on IMGT, with pseudogenes not included.

• MDAR checklist

### Data availability

All data and code used for the analysis in this manuscript are freely available online with no restrictions. All input FASTA sequences and code needed to recreate the analysis can be found via the AIMS GitHub page. Specific analysis for structural comparisons between interaction potentials and TCR-pMHC complexes are found via a separate repository, called PRESTO, also hosted on GitHub. Due to the significant time required to calculate the interaction scores calculated via AIMS, the calculated scores can be found on Zenodo. In case of future updates to either AIMS or PRESTO, the specific versions used for this manuscript are also hosted on Zenodo, as AIMS v0.8 and PRESTO v0.1.

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
