## [Editor Report]

This important work presents evidence that evolved biophysical compatibility between T cell receptors (TCRs) and MHC molecules is possible and a potential solution to the question of how TCRs could be biased towards MHC proteins given the massive diversity in both receptor and ligand. The evidence supporting the claims of the authors is solid, although the nature of these evolutionary questions makes it difficult to confidently answer some of the raised questions. The work will be of interest to immunologists, structural biologists, and evolutionary biologists.

---

## [Decision Letter]

[Editors' note: this paper was reviewed by Review Commons.]

**Decision letter after peer review:**

[Editors’ note: the authors submitted for reconsideration following the decision after peer review. What follows is the decision letter after the first round of review.]

Thank you for submitting the paper "Conserved Biophysical Compatibility Among the Highly Variable Germline-Encoded Regions Shapes TCR-MHC Interactions" for consideration by *eLife*. Your article has been reviewed by 3 peer reviewers, and the evaluation has been overseen by a Reviewing Editor and a Senior Editor. The following individuals involved in the review of your submission have agreed to reveal their identity: Eric Huseby (Reviewer #2); Brian Baker (Reviewer #3).

Comments to the Authors:

We are sorry to say that, after consultation with the reviewers, we have decided that this work will not be considered further for publication by *eLife*.

All reviewers agree that this study addresses an important topic, namely the sequence determinants of TCR-MHC binding modes. The sequence analysis in the study illustrates that the diversity of CDR loops and MHC contact surfaces is likely incompatible with hard-wired interaction motifs. However, the reviewers argue that the subsequent claims about the true origins of TCR:MHC docking orientations, and speculation about the origins of self-reactive TCRs, are based on flawed and unconvincing analyses. The reviews provide detailed suggestions as to how to improve the analyses to test the claims. We believe the substantial steps needed to address these reviews go beyond the scope of this manuscript but if the authors decide to expand on these suggestions, they can submit the manuscript as a new submission.

*Reviewer #1 (Recommendations for the authors):*

The authors investigate the origins of TCR:MHC docking orientation using information-theoretic sequence analysis simplified biophysical scoring, and inter-atomic contact analysis of solved TCR:pMHC ternary complexes. First, the authors show that the TCR CDR loops are more variable in sequence and in biophysical properties than the surface-exposed regions of the MHC. They conclude that "This mismatched diversity between the TCR germline-encoded CDR loops and the MHC α-helices suggests that conserved germline-encoded interactions are unlikely to exist for every possible molecular combination". Though not conclusive, this is consistent with the observed variability in the solved ternary structure databases, which show smoothly varying binding orientations rather than discrete recurring binding solutions. The authors also claim to see very little mutual information between TCR and MHC sequence positions, but they appear to be matching them up randomly (rather than using established TCR:MHC pairings from epitope-specific TCRs, for example, or TCRs from HLA-typed individuals), so it's not clear how this analysis could possibly find any covariation.

The heart of the study relies on a simplified 20x20 amino acid interaction matrix that is meant to capture basic biophysical interaction propensities. The sign of the values in the matrix is chosen to accord with intuition (opposite charges are favorable, like charges are unfavorable, etc), but the absolute values of the matrix values seem pretty arbitrary (all either 0, 0.5, 1, or 2). All matrix values for alanine and glycine are zero, despite their frequent involvement in tight hydrophobic packing interactions. The core calculation is to take all the residues in a given CDR1 loop (regardless of orientation: pointing toward MHC or toward the core of the TCR), and look up the interaction matrix scores for all surface residues (or maybe even all residues, period, it's hard to tell) of an MHC molecule, and sum up all the interaction scores. This single number (averaged over all HLA alleles) then reports the "interaction propensity" for that CDR1 loop sequence; if it's negative, then the loop/V gene has "severely limited interaction potential with HLA molecules". Despite the obvious problems with this – that the matrix is crude and arbitrary, that the sum involves many pairs of residues that couldn't possibly interact, etc, etc – the authors take these summed interaction scores as the basis for subsequent conclusions. For example, Figure 4 shows that TRBV7-2 and 7-3 have limited interaction potential (which appears to be related to them having glycine and alanine in their loops) with class I MHC; this finding is linked by the authors to the fact that these V genes are enriched in certain epitope-specific responses in celiac disease and multiple sclerosis, despite these enrichments being found in CD4 T cells. I looked at several TRBV7-2/7-3 containing ternary complexes (4mji, 5eu6, 5d2l, 4grl, 4ozh) and in fact, the TRBV7 segments are making extensive MHC contacts, dominating the TRAV segments in every case (4grl is a great example). It seems doubtful that the interaction scores derived from all pairwise residue matrix values are telling us anything about the intrinsic binding properties of the TCR V genes.

Next is a section entitled "Structural Data Validate Interaction Scores", in which they analyze atomic contacts in ternary complexes. Figure 5A certainly looks impressive at first glance, with tall bars for "predicted binders" and short or non-existent bars for "predicted NonBinders". But the problem is that there is no correction for the number of "nonbinder" V genes, and for example for CDR1A there only appear to be only 2 or 3 (Figure 4B), which may or may not have contributed to the database of solved structures. Thus the preponderance of observed contacts coming from "predicted binders" could just be due to the structural database composition, with binder and non-binder V genes making interactions at the same rate. The other problem with this analysis is that the contact analysis itself is flawed: the distance threshold is too small for hydrophobic interactions (4.5A would be better); there are too few total contacts being found (average of 1.35 per structure) the atom types included (referring here to the jupyter notebook https://github.com/ctboughter/PRESTO/blob/main/AIMS_interact_compare.ipynb) don't look right, since oxygen-oxygen and nitrogen-nitrogen can both form an H-bond donor-acceptor pair, and there's no evidence that hydrogens are being added to the structure; and the rules for counting "productive" contacts are too prescriptive (no carbon-carbon hydrophobic contacts allowed between polar or charged residues, even arg and lys with their long side chains). This latter has the consequence that the comparison to interaction scores becomes a little circular because the contact counting is driven by the same simplified biochemical intuition embodied in the pairwise interaction matrix. Much better would be to combine unbiased contact analysis (including backbone atoms) with an orthogonal measure such as buried surface area, and then look to see if predicted non-binder V genes really do make fewer interactions with MHC.

The remainder of the manuscript uses these interaction-matrix sums to investigate the determinants of the TCR:MHC docking mode. This is just not convincing, for the reasons outlined above, and also because there appear to be logical inconsistencies here. The concept is that MHC surfaces of "low interaction potential" (ie, alanine and glycine) define guardrails that limit the binding mode. Figure 8E has a nice cartoon showing the central ala/gly region in the class I alpha2 helix. The problem is that there are actually contacts throughout and on both sides of that "guardrail", which can be seen from Figure 7C, lower panel (161,164,165) or from a cursory examination of a few ternary complex structures. It's also not clear why, for class I α helix 2, MHC positions 143 and 144 have such low interaction scores (Figure 7C, upper panel) when in the alignment in Figure 8C those positions look similar to other R/K-containing positions.

On the positive side, the authors make their analysis scripts and notebooks very easily accessible, which is a big plus for reproducibility and transparency.

A few additional comments:

"The exposed residues on the α2-helix of HLA class I molecules are enriched in alanine and glycine relative to the α1- helix, which is highly unlikely to be involved in a specific, orientation-altering productive interaction" – alanine and glycine can be involved in highly specific packing interactions. Glycine, for example, can create pockets into which other side chains fit.

"Every crystallized TCR-HLA class II complex solved thus far adopts the canonical docking orientation whereby the TCR β-chain binds to the HLA α-chain helix, while the TCR α-chain binds to the HLA β-chain helix" – this is not correct, see 4y19 and 4y1a from the Rossjohn group.

"…obviating the need for a more precise approach" – do you mean "highlighting"?

This part of the methods is super-confusing (and I couldn't find it in the code): "Further, given that any single pair of amino acids on adjacent interfaces of protein binding partners can potentially form strong interactions without being meaningful for the formation of a given complex, we require that any productive interaction include a triad of at least weakly interacting residues".

As mentioned above, the whole mutual information analysis seems bonkers. How could there be any mutual information if the pairing between TCR and MHC is random/arbitrary? Please explain this part better:

"…every TCR should interact with every HLA allele. Humans largely possess the same TRAV and TRBV alleles, but each individual possesses a maximum of 12 HLA alleles. We expect that specific alleles that are unable to enforce the supposed evolutionary rules for canonical docking will not be allowed to persist in the population. Continuing from this assumption we then subsample the data and calculate the mutual information on this subsampled dataset. Each TRAV and TRBV allele (the input) is matched with a single HLA allele (the output), and the mutual information is calculated for these pairings".

*Reviewer #2 (Recommendations for the authors):*

The authors take an all-encompassing computational approach to analyzing TCR CDR – MHC interactions with the goal of identifying repetitive use of complementary protein-protein interaction events. On a first pass, there does not appear to be a significant contribution (to the T cell repertoire) of truly conserved pairwise interactions that drive MHC restriction. In contrast, their 'whole repertoire-wide approach' strongly supports the general concept that TCRs find opportunistic ways to bind pMHC using biochemically similar interactions.

First, I want to state that I really enjoyed reading this paper. I think it is written very well, which is quite important for papers on this topic as it can be a struggle even for seasoned immunologists to comprehend how there might be 'rules of engagement' when both the TCR and MHC/HLA are highly variable proteins. My comments will largely focus on issues that may help the authors provide the readers with a better understanding of the background and what their program does, and does not do. I will use statements within the manuscript to highlight these challenges.

In the abstract, "The formation of the TCR-peptide-MHC complex (TCR-pMHC) can be broken into two types of interactions, one between the hypervariable TCR CDR3α/β loops and the presented peptide and the second between germline-encoded regions of the TCR and MHC. "

– This is not an accurate statement. There are significant interactions between the CDR3 and MHC, as well as CDR1 and peptides. E.g., CDR1a often engages p-1 and p2 peptide residues, CDR3b almost always engages at some level, MHC-IIa61 area, and CDR3a with MHC-IIb 60area. Within the manuscript, the authors back off a bit from their hyper-simplistic statement, however, having such a blunt untrue statement in the abstract is not reasonable.

"Instead, binding properties such as the docking orientation is defined by regions of biophysical compatibility between these loops and the MHC surface."

– The authors spend a lot of effort working through certain variables that contribute to the binding reaction. I am wondering if the authors took account of shape complementarity (e.g., PMID: 9628472) and CDR loops that carry different types of conserved canonical structures (e.g., PMID: 10656805). One could imagine that based on the protein folding requirements of CDR regions, certain residues are in the interface whereas others are internal to the CDR structure and cannot actually contribute directly to binding.

"We selected a key hydrogen bonding network in the KIR2DL2-HLA-C*07:02 interface [50] and compared this to the evolutionarily conserved YXY motif of CDR2β [19, 21]".

– This is a good example to discuss the point above. it is important to know structurally, where each residue is. For example, the first Y (46 or 48 depending upon the nomenclature, above) often does not directly contribute to pMHC binding but may be important for the "outline structure" of the CDR loop itself. In addition, the authors do not discuss Van der Waals interactions really at all. Much of the TCR-pMHC interface (binding affinity) is driven by the exclusion of water, a property that is very difficult to assess on an amino acid-amino acid pairwise allotment of interaction energy. I was hoping that once the authors started to discuss "areas of binding potential" the contribution of non-side chain to side chain interactions would be discussed. It is unclear to this reviewer if these types of interactions are accounted for within their algorithm or if they are largely ignored.

In discussing the interaction potential, of amino acids, the authors cite and discuss a single manuscript.

42. P. Nandigrami, F. Szczepaniak, C.T. Boughter, F. Dehez, C. Chipot, and B. Roux. Computational assessment of protein-protein binding specificity within a family of synaptic surface receptors. Journal of Physical Chemistry B, 2022.

There is of course an empirical and computational field of study for how proteins bind one another as well as for TCRs and pMHC (e.g., PMID: 10410805, PMID: 16193038, PMID: 18946038, PMID: 27348411). Some more inclusive discussion of past ideas about how proteins interact with one another and whether old ideas remain accurate could add to the overall discussion.

"Figure 4: Interaction score between every TRBV (A) or TRAV (B) sequence and HLA allele for all four germline-encoded CDR loops. "

– Why are alanine and glycines assumed to be zero/non-interacting? Does a binding reaction care if a contact is a side chain-side chain, backbone-backbone, or mix? Indeed, when the authors "counted the contacts" I assume many of the side chains were indeed interacting with backbone atoms. It has also been suggested that some side chains can contribute negatively to interfaces (e.g., PMID: 17041605). Another question, perhaps for the algorithm used, does it take into account the frequency at which say X and Y amino acids actually occur at a possible site of interactions. It is mentioned that autoimmune-prone T cell repertoires are biased for certain TCR usage, does this bias include matching/non-matching HLA areas of recognition? There was some discussion on this but a clearer picture (if there is one) could be spelled out for the non-expert.

The interaction potentials also succeed in predicting TCR complexes that will not make contact with MHC. 20 of the 22 structures predicted to have poor CDR2β binding make no contact with MHC, while the last two only make one contact with MHC (Figure 5C). Further, all 8 structures predicted to have poor CDR1β binding make no contact with MHC. (Figure 5C). Again, this prediction accuracy is lower for class II predictions (Figure 5D).

– This is a super interesting idea that may unlock a lot of what is going on. One wonders how much of this is random chance, i.e., if a different TCR-pMHC with the same V genes and HLA would behave similarly. Also, do these structures preclude (or are driven by) CD1-peptide contacts, or are the structures carry such a different docking orientation as to completely preclude the CDR1 and CDR2 regions from being part of the binding interface?

The exposed residues on the α2-helix of HLA class I molecules are enriched in alanine and glycine relative to the α1- helix, which is highly unlikely to be involved in a specific, orientation-altering productive interaction.

– In practice, it is this reviewer's understanding that there are exit contributions of Van der Waals interactions at these sites. Indeed, early ideas suggested that the diagonal area of pMHC (MHCa 61, MHCb73) used this divot for shape complementary purposes.

It is important to note that these interaction potentials take an unbiased approach, calculating every possible interaction between TCR and MHC residues to produce this final score.

– It was unclear if the authors mean position by position, or did they weigh whether a residue was actually surface exposed and capable of being part of the binding interface.

productive side-chain interactions between CDR loops and the solvent-exposed residues of the MHC helices.

– There does seem to be an (over) emphasis on side-chain interactions. And less so on the clustered ability for VDW and/or inhibitory interaction.

– In general, there are quite a number of T cell development citations with actually very little discussed the role of thymic selection in and/or clonal T cell responses in skewing the TCR-pMHC interface to conform to selective pressures. E.g., TCRs can't be too good/cross-reactive or they would undergo central tolerance.

"In calculating the interaction score, we assume that productive contacts are only made by the side chains of the interacting residues. This simplification does not capture all TCR-pMHC complex contacts, but here we are looking for selectivity enforced by specific TCR-MHC interactions mediated by side-chains."

– Though stated as a caveat, perhaps some effort could be made to include side-chain to the backbone, etc interactions.

*Reviewer #3 (Recommendations for the authors):*

Boughter and Meier-Schellersheim describe an analysis of TCR-peptide/MHC complexes, aiming to gain an understanding of the underpinnings of the "common" TCR binding geometry. This is fundamental to understanding the MHC restriction of TCRs and how T cells scan and readout peptides. They begin with a comprehensive bioinformatics approach, move to a structural analysis to help interpret the informatics, and bring in biophysical computations. The overall conclusion that specific contacts between TCR genes and MHC proteins are not necessarily pre-programmed and that traditional TCR binding geometries emerge from biophysical compatibility is supported by the data and consistent with recent findings. In general, the work and the conclusions are an advance and place recent findings into perspective. However, the strength of evidence is weakened by choices made in characterizing structures, computing energies, and a strained reliance on "roles" played the parts of the interface which have been discounted many times yet persist in the literature. The latter in particular weakens the discussion and how the authors view the impact of their work.

The major strength of the paper is the approach taken; I found the comparative analysis of TCR and MHC genetic variability at the sequence level particularly compelling. Bringing in KIRs as a control was also a strong way to support the arguments. There is one major technical weakness in that, as far as is clear from the methods, interatomic interactions were considered with a 3.5 Å cutoff. This is woefully inadequate. Electrostatic interactions can be strong at long distances, which the authors really need to consider – say, going out to 6 Angstroms or so (there is much-published literature on short- and long-range electrostatics in protein interfaces). The importance of long-range electrostatics in TCR-peptide/MHC complexes has been demonstrated previously, particularly in prior work that aimed to address the same problem studied here. The authors also fall victim to the common immunology trope that CDR3-peptide interactions drive specificity, leaving CDR1/CDR2 to bind MHC proteins, i.e., the CDR loops have "roles" in binding. In the very first high-resolution structure of a TCR-peptide/MHC complex, CDR3 interactions with a class I MHC were noted and remarked on, as were CDR1 and CDR2 interactions with the peptide. Later work showed that these CDR3-MHC and CDR2-peptide interactions were critical for binding. These findings have been replicated several times now. The authors' introduction of this perspective of different loops of the TCR playing evolved roles (CDR3->peptide, CDR1/2->MHC), and their interpretation of their findings in light of it, weakens the papers' conclusions and impact, and it is a missed opportunity that can be addressed with the authors' approach.

The authors also should consider other literature for a greater impact on their work. For example, they also exclude backbone interactions – this is a curious omission from a biophysical perspective, and others in the field have published on the importance of backbone-mediated interactions (hydrogen bonds mostly) in stabilizing TCR interfaces. The authors also mention but fail to address T cell selection and the role of selection (and possibly coreceptor) in 'enforcing' what we get and have seen structurally (i.e., the idea that pre-selection TCRs bind all over the place, but selection ensures we get ones that bind right and work). Much has been written about this and it should be included.

1) The very first high-resolution crystal structure of a TCR-pMHC complex by Garboczi and Wiley in the 90s (PMID 8906788) showed CDR3 contacts to the MHC and germline CDR1/2 contacts to the MHC. Later biophysical studies by our own group showed these were crucial for binding (PMID 23736024). Other work has shown the same. Thus, although it is common to say that diverse CDR3 loops bind peptide and germline-encoded CDR1/2 loops bind the MHC, this is not supported at the atomic or energetic level. It actually plays INTO the authors' argument about opportunism/compatibility, but curiously the authors do not discuss it. They should. These observations and the idea that "roles" are not hardcoded into the TCR CDR loops play right into the authors' opportunistic argument introduced at the end of the paper.

2) A 3.5 Å cutoff is far too limited and ignores long-range electrostatics. Our own work addressing the same problem (which also introduced the notion of opportunism/compatibility) found signals for some "sloppy" evolved compatibility but only if we moved to longer ranges (PMID 26884163). The authors should re-evaluate their energetic analysis using longer-range cutoffs. To avoid greatly complicating the analysis, longer ranges could be done only with charged side chains. It was also very curious to omit main chain interactions, something which the authors might want to work back in (see PMID 17041605).

3) The authors should really address the question of how thymic education influences what we see. For example, we recently published a TCR that binds with an outlier geometry (not reverse) which signals just fine – an example of a class-mismatched TCR (emerged from a CD4^+^ T cell but binds a class I). This TCR is a bit weird in that it has an unusually long CDR3b loop that contacts both peptide and MHC (point 1 again). We also concluded that this is a weird TCR that somehow escaped normal thymic selection, implying that maybe the pre-selection repertoire has TCRs that bind crazily and one role of thymic selection is to filter these, giving us TCRs that are somehow "better" biologically (maybe they signal better, or possess lower x-reactivity, etc.). The authors need to work this thinking in. Relevant papers are PMID 36424374 and PMID 30833553.

4) The authors use "compatibility" and "opportunistic" to describe TCR binding from a biophysical perspective, contrasting this with the hard-coded model. These are not new concepts though, and although the authors have greatly expanded on the topic (albeit with the limitations above), they should make note of this. They do reference some of the appropriate literature, but clarifying how they are expanding on the topic would strengthen the impact of the work.

[Editors’ note: further revisions were suggested and these were then sufficiently addressed prior to acceptance, as described below.]

Thank you for resubmitting your work entitled "Conserved Biophysical Compatibility Among the Highly Variable Germline-Encoded Regions Shapes TCR-MHC Interactions" for further consideration by *eLife*. Your revised article has been evaluated by Tadatsugu Taniguchi (Senior Editor) and a Reviewing Editor.

The manuscript has been improved but there are some remaining issues that need to be addressed, as outlined below:

Essential revisions:

As you can see from the report, the reviewers appreciate the changes done for revision. After an extensive discussion, the overall consensus of the reviewers is that while the concept of evolved biophysical compatibility is possible and a potential solution to the question of how TCRs could be biased towards MHC proteins given the massive diversity in both receptor and ligand, it is a concept that is exceptionally difficult to demonstrate and the paper still has some wishful thinking. For this manuscript to move forward, we request that you tone down the paper, remove the claims highlighted by reviewer #1, and present the concept as an interesting possibility for which some evidence is offered but no solid proof (see report from reviewer #1 for details).

We also had a discussion with regards to the suggestion of reviewer #2 to perform a similar analysis on BCRs to verify that the signal is not spurious. We acknowledge that this might be beyond the scope of the current paper. However, if the authors chose to do this analysis, it can help solidify some of the claims.

*Reviewer #1 (Recommendations for the authors):*

I recognize the time and effort that the authors have invested in responding to the reviews of the first version of the manuscript. It is appreciated that they recognized the circularity of the original Figure 5 and removed it, adjusted the distance thresholds and sequence-filters for contacts analysis, and that they have also removed references to the origin of self-reactive TCRs.

My concerns with regard to the claims about V-gene interaction potential and determinants of the binding mode still stand, since the relevant text hasn't been modified and the author's responses are not convincing. For example, the detailed analysis of the TRBV7-2 containing complexes provided by the authors in the response appears to disprove the AIMS-based prediction that this gene has low interaction potential: "Certainly PRESTO agrees with these structural interpretations, suggesting CDR2B dominates the germline interactions here, with 13/15 SC-SC contacts." The contorted logic that the authors produce to explain this disconnect doesn't really make sense: "However, yet again we have an abnormally high number of CDR2B backbone-backbone interactions, 14, suggestive of nonspecific tight packing not driven by TCRB specific interactions". What exactly is "nonspecific tight packing"?

The authors also continue to over-sell their findings in the newly introduced text. For example, in describing the new Figure 5, the authors state: "This comparison shows exceptional agreement between our bioinformatic results and structural analyses". But when one compares Figure 5a and 5b, for example, the agreement is pretty dubious. And in 5d, *none* of the differences are significant, and many show the wrong directionality, for example, the median value for "Weak TRBV" is always greater than or equal to the median value for "Moderate TRBV". And in the new text describing the AIMS potential: "The AIMS interaction potential, which can swiftly analyze thousands of sequences, has significantly outperformed more physically detailed and computationally expensive models. In a binary classification of a large database of structurally similar protein complexes, the AIMS interaction potential was capable of distinguishing binders and non-binders to an accuracy of 80%, whereas calculations run on over 45µs of simulated all atom trajectories could only distinguish to an accuracy of 50%. " I looked back at this reference, and what the authors neglect to mention is that the 80% performance comes from a highly parameterized model based on a linear discriminant analysis fitting a weight for each pair of residues in the interface-- it's not at all analogous to the calculation here in which AIMS scores are directly summed up. It's also a single family of interacting proteins.

*Reviewer #2 (Recommendations for the authors):*

With regards to the manuscript in general, in some places, the authors seem to want to have their cake and eat it too. Particularly, the idea that TCRs are evolutionarily biased to recognize MHC, included stating support for the "codon model" while at others suggesting that CDR1s and CDR2s have only minimal (complementary) roles in binding. With the extension suggesting that some TRAVs and TRBVs have no (or very minimal) MHC/HLA binding potential. This later argument would suggest that antibodies, fully capable of creating diverse CDR3s, should similarly have a (modest, strong) ability to bind pMHC ligands. I suppose a computational test of the general idea the authors are putting forward would be to use their AIMs platform with human antibody CDR1s and CDR2s to see if these were all net no-binding or negative binding with MHC. However, I do not like the idea of bringing up additional questions/tests of the model during a re-review.

---

## [Author Response]

[Editors’ note: the authors resubmitted a revised version of the paper for consideration. What follows is the authors’ response to the first round of review.]

Comments to the Authors:We are sorry to say that, after consultation with the reviewers, we have decided that this work will not be considered further for publication by eLife.All reviewers agree that this study addresses an important topic, namely the sequence determinants of TCR-MHC binding modes. The sequence analysis in the study illustrates that the diversity of CDR loops and MHC contact surfaces is likely incompatible with hard-wired interaction motifs. However, the reviewers argue that the subsequent claims about the true origins of TCR:MHC docking orientations, and speculation about the origins of self-reactive TCRs, are based on flawed and unconvincing analyses. The reviews provide detailed suggestions as to how to improve the analyses to test the claims. We believe the substantial steps needed to address these reviews go beyond the scope of this manuscript but if the authors decide to expand on these suggestions, they can submit the manuscript as a new submission.

While we will go through the reviewer comments point-by-point, we believe it would be helpful to summarize our most important changes beforehand, and reference these throughout the point-by-point comments. The concerns of the reviewers were largely focused on similar features of the first submission of this manuscript.

Issue 1: The AIMS interaction matrix is untested, arbitrary, or has not been sufficiently compared to other methods in the literature.

To address the relatively arbitrary assignment of absolute values in the AIMS interaction matrix, the analysis has been repeated with a 3-value matrix, which we will refer to as V0. Interactions are either deemed negative (-1), neutral (0), or positive (+1). Due to the large number of supplemental figures already included, we include these V0 figures in the end of this review (Author response images 1-4). We find that overall, our observations remain the same regardless of the interaction matrix used (V0 or the original matrix, which we will refer to as V2). This is in line with previous work taking a similar interaction potential approach, where the results are largely unchanged by minute differences in the potential used [Kosmrlj et al. PNAS 2008. PMID: 18946038].

**Author response image 1. sa2fig1:** Class I Interaction Scoring with v0 Matrix (Compare to Figure 4).

**Author response image 2. sa2fig2:** Class IIa Interaction Scoring matrix with V0 scoring (Compare to Figure S8A).

**Author response image 3. sa2fig3:** Class IIb interaction scoring matrix with V0 scoring (Compare to Figure S8B).

**Author response image 4. sa2fig4:** The V0 scoring matrix (for direct comparison with Supplemental Figure 9A).

In addition to the binarized form of the matrix, it should be noted that the AIMS interaction matrix significantly outperforms methods that attempt to capture more details of the interactions. Specifically, looking at the ability of various algorithms to predict interacting or non-interacting molecules (Figures 7, 8, 9, and 10 in the cited Nandigrami et al. manuscript) AIMS distinguishes between known binders or non-binders with a significantly higher accuracy compared to calculations based on all-atom molecular dynamics simulations starting from crystals or modelled structures. In other words, the software outperforms more physically detailed models in predicting protein-protein interactions, the exact problem we are interested in at present.

A paragraph dedicated to the high performance of the AIMS interaction matrix approach is now included in the main text, lines 212-221**:**

“The AIMS interaction potential, which can swiftly analyze thousands of sequences, has significantly outperformed more physically detailed and computationally expensive models. In a binary classification of a large database of structurally similar protein complexes, the AIMS interaction potential was capable of distinguishing binders and non-binders to an accuracy of 80%, whereas calculations run on over 45us of simulated all-atom trajectories could only distinguish to an accuracy of 50%. While not as biophysically descriptive as methods such as alchemical free energy perturbation [Gumbart 2013a] or potential of mean force-based calculations [Gumbart 2013b], the AIMS interaction potential generates accurate predictions for problems that are intractable with current limitations to computation due to the number of protein complexes of interest.”

Issue 2: Closely related to Issue 1, the scoring of alanine and glycine interactions as “0” in the interaction matrix was largely deemed incorrect by the reviewers, due to the ability of these amino acids to be involved in “tight packing” hydrophobic interactions, or as hydrogen bond partners using backbone atoms.

In this response, we will discuss only the side chain interactions of glycine and alanine. For a discussion on backbone hydrogen bonding (with sidechains or other backbones), see issue/response 5.

The decision to refer to alanine and glycine interactions as “0” in the matrix is largely due to their lack of sidechain hydrogen-bonding capability and relatively limited (de)solvation penalty. With their minimal sidechains, the energetic benefit of a hydrophobic interaction is likewise minimal. Whether one considers the hydrophobic effect as driven by ordered waters or disruption of water-water hydrogen bonds, the smaller hydrophobic volume of Ala and Gly will have a more limited entropic or enthalpic effect. The smaller the hydrophobic volume, the smaller the effect of the singular hydrophobic molecule [Chandler *Nature Insight Review* 2005. PMID: 16193038].

We further quantitatively confirm that such interactions are rare in TCR:pMHC complexes crystallized thus far (see new Figure 5). Of note, this new figure appears to show 1. That, in fact, glycine and alanine rarely make sidechain-sidechain (SC-SC) contacts and 2. That the qualitative agreement between the AIMS interaction potential and these counted contacts further supports the use of the interaction potential in this study (see issue/response 1). It is important to note these figures are generated using the new structural cutoffs discussed in issue/response 3.

There is an exception to the lack of SC-SC interactions involving Ala or Gly, with the only enriched interaction coming from Tyr-Ala interactions. However, it is important to note that such interactions have been the center of intense study [Feng et al. *Nat. Immuno* 2007; Garcia et al. *Nat Immuno* 2008; and others] and may potentially be biasing the PDB. Further, while this interaction may genuinely be important for immune recognition, not every TRAV/TRBV gene includes tyrosine in the CDR loops. 9/45 TRAV and 18/48 TRBV alleles do not contain a single tyrosine.

Upon further consideration of these “tight hydrophobic packings” that have been observed between Tyrosine and Alanine, it is important to note that such packing (as in the “knob-in-hole” interaction) is unlikely to be long-lived in many structures. Side chains are inherently motile, and shifting of these side chains is likely to occur over the course of the formation of an interaction interface. More stable are hydrogen bonds, charged interactions, and stronger hydrophobic interactions.

To support this dynamic interpretation of side chain interactions, we now include a paragraph in the discussion dedicated to a truly dynamic interpretation of TCR-pMHC interactions. We note that in triplicate all-atom molecular dynamics trajectories of PDB 1FYT, the tyrosine-alanine “tight packing” lasts only between 10-20ns across these triplicate trajectories (Figure 9, Supplemental Figure S14), suggesting again that this interaction is less key to the interface, and more an opportunistic placement for this side chain. We further include new text in the discussion focused on how such a dynamic interpretation has been considered previously, and what these new results mean for TCR-pMHC interactions.

Issue 3: The cutoffs used in the analysis of previously published structures were improperly estimated, and the consideration of only “productive” contacts artificially boosted the agreement between the AIMS scoring and structural data.

The structure parser suggestions have now been implemented per the comments of Reviewers 1 and 3, with hydrophobic interactions of 4.5Å, charged interactions of 6Å, and a more permissive definition of “productive” interactions (i.e. C-C packing interactions between hydrophobic and hydrophilic residues counted, O-O hydrogen bonds allowed if Ser, Tyr, and Thr are involved, N-N allowed if His involved). We also now include all interactions with the protein backbone, and discuss these interactions more in the text.

We agree with reviewer 1 that upon re-inspection of Figure 5 and the way a “productive contact” was defined, the original structural validation can be seen as somewhat circular. As such, we have removed the original figure 5 and instead replaced it with results that:

1. Further validate the form of the AIMS interaction matrix itself (panels A, B) and

2. Provide a more robust, quantitative structural validation of the AIMS interaction matrix (panels C, D).

3. Issues regarding the normalization between groups and the over/underrepresentation of certain weak or strong binders have also been addressed, with more robust statistical tests included to determine significance in differences between groups, rather than simple bar plots.

Issue 4: The treatment of the TCR:pMHC interaction as mediated exclusively by germline-germline or CDR3-peptide interactions does not reflect the reality of the interface. CDR3 can frequently contact germline regions, and germline CDR loops can frequently contact peptide.

Originally these statements were written from the theoretical sense “it is useful to approximate this interaction as one between germline-encoded regions or hypervariable regions”. This is obviously not an accurate reflection of reality. However, upon re-reading the manuscript with the reviewers’ comments in mind, we find that our intent was not clearly reflected in the text.

We have significantly updated the abstract and main text to more properly reflect our assessment of these interactions. It is important to note, however, that our analysis still explicitly ignores CDR3 and peptide interactions that would be well beyond the scope of the current manuscript. Our goal here is to consider the TCR-pMHC interface in a reductionist approach and consider just the regions that are unchanging across all complexes. In almost all structures solved thus far, the germline CDR1/2 loops interact with the MHC α helices. So, our goal is to determine rules for how might happen for the “average” TCR-pMHC complex, while realizing that there will be strong deviations from this average complex. Hence why we make a point to discuss these outliers at length (lines 34-45):

“Importantly, previous work has found a large list of deviations from these classic "rules of engagement". The hypervariable CDR3 residues can dominate the interactions with the germline-encoded MHC α-helices [Piepenbrink 2013, Singh 2022], germline encoded CDRs can interact strongly with peptide [Piepenbrink 2013], an exceptionally long peptide can "bulge" causing separation between germline regions [Tynan 2005], or the docking orientation can even be entirely reversed [Gras 2016, Zareie 2021]. When considering the large number of possible TCR-pMHC interactions and the substantial cross reactivity of TCRs [Riley 2018, Sewell 2012], these historically non-canonical interactions may in fact be rather common.”

Issue 5: The analysis presented in this manuscript, specifically the AIMS interaction matrix, does not account for various structural features, including shape complementarity, backbone interactions, and relative solvation/desolvation of certain sidechains.

We understand and appreciate the vast structural and biochemical literature that has contributed to our understanding of TCR-pMHC interactions thus far. A complete understanding of how a TCR discriminates between self, non-self, or altered-self peptides will require understanding the precise contributions of these structural features, and some not mentioned by the reviewers (catch bonds, dwell time, dynamic matching, etc). Unfortunately, the explicit incorporation of such structural, energetic, and dynamic features is outside of the scope of the AIMS analysis, and to our knowledge, well beyond the abilities of any modeling approach. We now explicitly address what our manuscript does and does not attempt to do in the discussion (lines 511-528) and elaborate further below:

“The ideal approach for studying TCR-pMHC interactions would involve generating a comprehensive set of crystallized structures, exhaustive biochemical experiments to pinpoint contributions to binding affinity and kinetics, and activation assays to thoroughly understand the nuances that underlie complex formation. While such thorough efforts have been undertaken to understand structural strategies of binding to HLA-A2 [Blevins 2016], the potential evolutionary conservation of TRBV8-2 in binding MHC molecules [Scott-Browne 2009, Scott-Browne 2011, Garcia 2012, Feng 2007], and the by residue contributions of binding of the A6 TCR [Piepenbrink2013], the diversity inherent to the TCR-pMHC interaction makes such efforts impossible to scale across all possible binding partners.

The work presented here cannot replace these comprehensive experimental techniques. However, it can provide a good first estimate of how well given TCR-MHC pairs may bind. Absent rich experimental data on major parts of the TCR and MHC repertoire space, can we attempt to approximate how a fictitious TCR-MHC interaction would occur absent peptide and CDR3? Comparisons of our computational results to experiment suggest that yes, in fact, we can generate some strong approximations to build off of, with clear deviations from these predictions largely driven by outlier structures. As we continue to build these interaction potentials and the AIMS software as a whole, we hope to continually add modular improvements to consider how these initial germline interaction assumptions are altered by peptide and CDR3 to build more predictive tools for interactions and binding.”

In short, crystallography and biochemistry are the gold standard for understanding the many of nuances of TCR-pMHC interactions. Structural modeling could provide insights, but thus far has proven unreliable for adaptive immune interactions [Evans et al. 2022 https://doi.org/10.1101/2021.10.04.463034]. Likewise, structure-free machine learning approaches have largely failed to be predictive outside of narrow parameter spaces [Montemurro et al. 2021. PMID: 34508155; Jokinen et al. 2022. PMID: 36477794; Meysman et al. 2023. https://doi.org/10.1016/j.immuno.2023.100024 ]. Our goal is to explore how well a reductionist model that encodes some degree of physical realism can generate explanations for experimental observations made thus far.

Inclusion of phenomena such as specific sidechain-main chain interactions or shape complementarity would imply knowledge of the structures of interest, which is grossly overestimating the scope of our knowledge of these interactions thus far. Referring specifically to the percentages of complex space crystallized to date (lines 70-78)**,** we simply don’t have the data to extrapolate out to all possible complexes. Absent these data, we would need to generate strong assumptions about the presumed structural interactions between the TCR and the MHC, which are going to be strongly influenced by the CDR3, the peptide, the TRAV/TRBV pairing, and the MHC involved in the interaction.

Despite the lack of inherently structure-dependent interactions in AIMS, the newest version of the manuscript now more explicitly counts SC-backbone interactions in our structural analysis (Supplemental Figure S10). While we could potentially generate an empirical SC-backbone interaction matrix from these values, we find that the contact maps are too strongly biased by the structures crystallized thus far.

Reviewer #1 (Recommendations for the authors):The authors investigate the origins of TCR:MHC docking orientation using information-theoretic sequence analysis simplified biophysical scoring, and inter-atomic contact analysis of solved TCR:pMHC ternary complexes. First, the authors show that the TCR CDR loops are more variable in sequence and in biophysical properties than the surface-exposed regions of the MHC. They conclude that "This mismatched diversity between the TCR germline-encoded CDR loops and the MHC α-helices suggests that conserved germline-encoded interactions are unlikely to exist for every possible molecular combination". Though not conclusive, this is consistent with the observed variability in the solved ternary structure databases, which show smoothly varying binding orientations rather than discrete recurring binding solutions. The authors also claim to see very little mutual information between TCR and MHC sequence positions, but they appear to be matching them up randomly (rather than using established TCR:MHC pairings from epitope-specific TCRs, for example, or TCRs from HLA-typed individuals), so it's not clear how this analysis could possibly find any covariation.1A.The heart of the study relies on a simplified 20x20 amino acid interaction matrix that is meant to capture basic biophysical interaction propensities. The sign of the values in the matrix is chosen to accord with intuition (opposite charges are favorable, like charges are unfavorable, etc), but the absolute values of the matrix values seem pretty arbitrary (all either 0, 0.5, 1, or 2).

See response 1.

1B. All matrix values for alanine and glycine are zero, despite their frequent involvement in tight hydrophobic packing interactions.

See response 2.

1C. The core calculation is to take all the residues in a given CDR loop (regardless of orientation: pointing toward MHC or toward the core of the TCR), and look up the interaction matrix scores for all surface residues (or maybe even all residues, period, it's hard to tell) of an MHC molecule, and sum up all the interaction scores. This single number (averaged over all HLA alleles) then reports the "interaction propensity" for that CDR1 loop sequence; if it's negative, then the loop/V gene has "severely limited interaction potential with HLA molecules". Despite the obvious problems with this – that the matrix is crude and arbitrary, that the sum involves many pairs of residues that couldn't possibly interact, etc, etc – the authors take these summed interaction scores as the basis for subsequent conclusions.

Due to multiple confusions regarding the calculation of the interaction scoring, a more precise description has been provided in the methods (lines 604-614). We apologize that the initial explanations were not sufficiently clear.

Regarding the “arbitrariness” of the matrix, see response 1.

Lastly, regarding the pairs of residues that couldn’t possibly interact, the software is capable of extending or limiting the scope of the amino acids involved in the calculation. In the main text, the entirety of the TCR-accessible residues on both MHC α-helices are included in the calculation, to reflect the fact that, a priori, we don’t have a reason to expect any given TCR to bind in the “canonical” conformation. However, including just amino acids on the “proper” α helix does not strongly change the calculations (Author response image 5).

**Author response image 5. sa2fig5:** By-gene interaction scores with “proper” helix interactions (i.e. TRAV interactions with Class I alpha2 helix, TRBV interactions with Class I alpha1 helix).

**2.** For example, Figure 4 shows that TRBV7-2 and 7-3 have limited interaction potential (which appears to be related to them having glycine and alanine in their loops) with class I MHC; this finding is linked by the authors to the fact that these V genes are enriched in certain epitope-specific responses in celiac disease and multiple sclerosis, despite these enrichments being found in CD4 T cells. I looked at several TRBV7-2/7-3 containing ternary complexes (4mji, 5eu6, 5d2l, 4grl, 4ozh) and in fact, the TRBV7 segments are making extensive MHC contacts, dominating the TRAV segments in every case (4grl is a great example). It seems doubtful that the interaction scores derived from all pairwise residue matrix values are telling us anything about the intrinsic binding properties of the TCR V genes.

We thank the reviewer for pointing out the mistake made in the text. We were well aware that the TRBV7 enrichments were found in CD4^+^ T cells, and instead meant to point readers to the interaction matrix with Class II MHC found in Supplemental Figure S8. This has now been fixed in the text.

We hope that our previous responses to reviewers clear up what the interaction scoring matrix is telling us about the intrinsic binding properties of the TCR V-genes. Further, we greatly appreciate the list of specific PDBs to examine to back up our claims to an even greater extent. We can go through point by point and discuss the binding mechanisms of these example PDBs using structural visualization and the PRESTO software with new interaction cutoff settings.

5EU6 [Altered Self Antigen] – At first glance, CDR2B does appear to dominate the interface [17/26 SC-SC contacts], although these appear to simply be an artifact due to tight packing of CDR2B, which makes 8 backbone-backbone contacts according to our new, permissive interaction cutoffs. This is far above the median number of backbone backbone contacts in our analysis (see Figure 5D). A closer inspection of the structure suggests this tight packing is a due to CDR3B dominating the interface, both with peptide and with MHC. This in turn forces the germline CDR loops into closer contact with MHC. Details of the contacts made by CDR2B confirm this loop is unlikely responsible for tight packing, as the majority of contacts involve nonspecific packing interactions between hydrophobic and hydrophilic residues near the edge of the interaction cutoff [all above 4Å, near the more generous packing cutoff]. More specific interactions (salt bridges) are formed between TCRB and MHC, but these are TCR framework interactions, again due to the biased docking angle generated by CDR3B.

5D2L [CMV Antigen] – Not dominated by TCRB, only 8/36 SC-SC contacts made by TCRB. There are some interactions between CDR2B and MHC sidechains and backbone atoms, but these are limited. Further, these non-sidechain interactions are not meant to be covered by the AIMS interaction potential, as discussed elsewhere in the manuscript and in these review responses. Further, we should expect by the AIMS interaction potential that TRAV contributes just slightly more contacts than TRBV, as TRAV24 is likewise binned on the lowest end of the “moderately interacting” TRAV genes. We would predict, as we find here, that CDR3 would need to dominate the interface.

4MJI [HIV Antigen] – Not dominated by TCRB, only 16/45 SC-SC contacts made by TCRB. Of these 16, 10 are nonspecific packing C-C interactions between hydrophilic amino acids. Again, the dominance of TCRA is consistent with the predictive power of the AIMS interaction potential, as TRAV17 is a predicted strong binder. Indeed, we see multiple strong, specific interactions [ILE-LEU hydrophobic interaction, THR-ARG Hbond, ASN-GLN Hbond, ARG-GLU Salt Bridge, ARG-GLN Hbond, ASN-ARG Hbond]. Suggesting the AIMS interaction potential’s exceptional ability to provide insights into the intrinsic binding properties of TCR V genes.

40ZH [Celiac Autoimmune] – While a cursory look at the structure or the PRESTO structure parser may suggest a strong involvement of TCRB [10/23 SC-SC contacts], the original study provides further context [PMID 24777060]. The authors find that while there is skewed TRBV7-2 usage in DQ2.5-glia-α2 reactive TCRs, CDR1B and CDR2B only contribute a combined 14% to the overall buried surface area. Further, mutagenesis of what appear to be key contacting residues in CDR1B and CDR2B were insufficient to abrogate binding. Contrast this with mutations in TRAV germline regions or CDR3 loops, which did result in undetectable binding. Consistent with TRBV7-2 as an inherently weak interacting germline gene.

4GRL [MS Autoimmune] – Certainly PRESTO agrees with these structural interpretations, suggesting CDR2B dominates the germline interactions here, with 13/15 SC-SC contacts. However, yet again we have an abnormally high number of CDR2B backbone-backbone interactions, 14, suggestive of nonspecific tight packing not driven by TCRB specific interactions. Indeed, the original study [PMID 24136005] suggests that the titular dominating “single loop” is CDR3A. Suggesting that the CDR2B contacts are essentially bystanders in the binding process that happen to make contact with MHC due to the strong docking angle enforced by CDR3A.

Each of these structures offer important lessons that have been highlighted both throughout the manuscript and throughout this response to reviewers.

1. The CDR3 loops are capable of dominating interactions, bypassing every other “rule” imposed in this and other papers. In multiple cases, these CDR3-dominated interactions occur in the context of autoimmune events.

2. In considering a given TCR-pMHC complex, a distinction should likely be made between “contacts” and “interactions”, akin to the original PRESTO analysis that distinguished between hydrophobic-hydrophilic and hydrophobic-hydrophobic carboncarbon interactions. In the former (hydrophobic-phillic) case, which could be referred to as a simple “contact” there exists some VDW contribution to the binding energy, but potentially not a large enough contribution to overcome the de-solvation penalty of the hydrophilic residue. In the latter (hydrophobic-phobic) case, there is no such desolvation penalty for the hydrophobic residues, and this interaction can contribute strongly to the overall interface.

In the case of PDBs 40ZH and 4GRL, we see many such “contacts” made between MHC and TRBV7-2, but as is backed up by mutagenesis studies, these contacts contribute little to the overall binding free energy. Consistent with what the AIMS interaction potential suggests the intrinsic binding properties of TRBV7-2 should be.

3. Next is a section entitled "Structural Data Validate Interaction Scores", in which they analyze atomic contacts in ternary complexes. Figure 5A certainly looks impressive at first glance, with tall bars for "predicted binders" and short or non-existent bars for "predicted NonBinders". But the problem is that there is no correction for the number of "nonbinder" V genes, and for example for CDR1A there only appear to be only 2 or 3 (Figure 4B), which may or may not have contributed to the database of solved structures. Thus the preponderance of observed contacts coming from "predicted binders" could just be due to the structural database composition, with binder and non-binder V genes making interactions at the same rate. The other problem with this analysis is that the contact analysis itself is flawed: the distance threshold is too small for hydrophobic interactions (4.5A would be better); there are too few total contacts being found (average of 1.35 per structure) the atom types included (referring here to the jupyter notebook https://github.com/ctboughter/PRESTO/blob/main/AIMS_interact_compare.ipy nb) don't look right, since oxygen-oxygen and nitrogen-nitrogen can both form an Hbond donor-acceptor pair, and there's no evidence that hydrogens are being added to the structure; and the rules for counting "productive" contacts are too prescriptive (no carbon-carbon hydrophobic contacts allowed between polar or charged residues, even arg and lys with their long side chains). This latter has the consequence that the comparison to interaction scores becomes a little circular because the contact counting is driven by the same simplified biochemical intuition embodied in the pairwise interaction matrix. Much better would be to combine unbiased contact analysis (including backbone atoms) with an orthogonal measure such as buried surface area, and then look to see if predicted non-binder V genes really do make fewer interactions with MHC.

See response 3.

4. The remainder of the manuscript uses these interaction-matrix sums to investigate the determinants of the TCR:MHC docking mode. This is just not convincing, for the reasons outlined above, and also because there appear to be logical inconsistencies here. The concept is that MHC surfaces of "low interaction potential" (ie, alanine and glycine) define guardrails that limit the binding mode. Figure 8E has a nice cartoon showing the central ala/gly region in the class I alpha2 helix. The problem is that there are actually contacts throughout and on both sides of that "guardrail", which can be seen from Figure 7C, lower panel (161,164,165) or from a cursory examination of a few ternary complex structures. It's also not clear why, for class I α helix 2, MHC positions 143 and 144 have such low interaction scores (Figure 7C, upper panel) when in the alignment in Figure 8C those positions look similar to other R/K-containing positions.

We hope that we have largely addressed the “reasons outlined above” regarding how convincing our use and validation of the interaction matrix is.

We thank the author for their comments regarding Figure 8. The cartoon has been changed to more accurately reflect how the TCRs lay over the MHC, which is not as some singular shape laying over the surface, but rather as distinct loops contacting distinct portions of the MHC surface. These distinct loops reflect the results of Figure 7, but also were checked structurally (Author response image 6). Changes have also been made in the text to reflect that of course deviations to this general rule can be made. We just choose to show what is likely the most optimal configurational arrangement.

**Author response image 6. sa2fig6:** Structural information used to generate the cartoons of Figure 8. Overlay of a range of class I TCR-MHC structures with strong, moderate, and weak TRAV/TRBV AIMS-predicted binding propensity. While there is variation in the precise location of the CDR loops, they typically occupy a similar space over the class I MHC surface (colored to match the class I MHC in Figure 8E). We highlight with a dashed red circle where CDR loops tend to avoid docking directly over, likely due to the region of low interaction potential.

5. On the positive side, the authors make their analysis scripts and notebooks very easily accessible, which is a big plus for reproducibility and transparency.

We thank the reviewer for taking the time to go through the analysis scripts and notebooks. We hope that the changes we have made (with the help of the reviewer) will make the scripts and notebooks even more useful for the research community.

A few additional comments:6**.** "The exposed residues on the α2-helix of HLA class I molecules are enriched in alanine and glycine relative to the α1- helix, which is highly unlikely to be involved in a specific, orientation-altering productive interaction" – alanine and glycine can be involved in highly specific packing interactions. Glycine, for example, can create pockets into which other side chains fit.

See response 2.

7. "Every crystallized TCR-HLA class II complex solved thus far adopts the canonical docking orientation whereby the TCR β-chain binds to the HLA α-chain helix, while the TCR α-chain binds to the HLA β-chain helix" – this is not correct, see 4y19 and 4y1a from the Rossjohn group.

We appreciate the correction. This has been updated in the text.

8. "obviating the need for a more precise approach" – do you mean "highlighting"?

Yes, we have corrected this now.

9. This part of the methods is super-confusing (and I couldn't find it in the code): "Further, given that any single pair of amino acids on adjacent interfaces of protein binding partners can potentially form strong interactions without being meaningful for the formation of a given complex, we require that any productive interaction include a triad of at least weakly interacting residues".

We have changed the methods to hopefully clarify this confusion, and refer the reviewer to the updated methods section mentioned earlier in this response.

10. As mentioned above, the whole mutual information analysis seems bonkers. How could there be any mutual information if the pairing between TCR and MHC is random/arbitrary? Please explain this part better:"…every TCR should interact with every HLA allele. Humans largely possess the same TRAV and TRBV alleles, but each individual possesses a maximum of 12 HLA alleles. We expect that specific alleles that are unable to enforce the supposed evolutionary rules for canonical docking will not be allowed to persist in the population. Continuing from this assumption we then subsample the data and calculate the mutual information on this subsampled dataset. Each TRAV and TRBV allele (the input) is matched with a single HLA allele (the output), and the mutual information is calculated for these pairings".

We have now calculated this mutual information for crystallized complexes. However, the PDB must be considered a strongly biased source for these calculations. Of the 140 crystallized human class I structures, only 68 of these are unique in their

TRAV/TRBV/HLA usage. Further, as many as 91/140 involve HLA-A*02:01. All of these could potentially alter the mutual information in unpredictable ways that are not at all indicative of how these molecules were evolved.

Despite these caveats, new calculations of the mutual information using only those structures found in the PDB suggest that our initial approach to calculating the mutual information was accurate (Supplemental Figure S4). We see similar trends in the mutual information between CDR loops and the MHC α-helices, with no clear increase in information for interacting regions (TCRA-alpha2 and TCRB-alpha1). Instead, we see higher mutual information between the TRAV-TRBV pairings than TRAV-MHC or TRBVMHC.

Reviewer #2 (Recommendations for the authors):The authors take an all-encompassing computational approach to analyzing TCR CDR – MHC interactions with the goal of identifying repetitive use of complementary protein-protein interaction events. On a first pass, there does not appear to be a significant contribution (to the T cell repertoire) of truly conserved pairwise interactions that drive MHC restriction. In contrast, their 'whole repertoire-wide approach' strongly supports the general concept that TCRs find opportunistic ways to bind pMHC using biochemically similar interactions.1. First, I want to state that I really enjoyed reading this paper. I think it is written very well, which is quite important for papers on this topic as it can be a struggle even for seasoned immunologists to comprehend how there might be 'rules of engagement' when both the TCR and MHC/HLA are highly variable proteins. My comments will largely focus on issues that may help the authors provide the readers with a better understanding of the background and what their program does, and does not do. I will use statements within the manuscript to highlight these challenges.

We thank the reviewer for these encouraging comments.

2. In the abstract, "The formation of the TCR-peptide-MHC complex (TCR-pMHC) can be broken into two types of interactions, one between the hypervariable TCR CDR3α/β loops and the presented peptide and the second between germline-encoded regions of the TCR and MHC. "– This is not an accurate statement. There are significant interactions between the CDR3 and MHC, as well as CDR1 and peptides. E.g., CDR1a often engages p-1 and p2 peptide residues, CDR3b almost always engages at some level, MHC-IIa61 area, and CDR3a with MHC-IIb 60area. Within the manuscript, the authors back off a bit from their hyper-simplistic statement, however, having such a blunt untrue statement in the abstract is not reasonable.

See response 4.

3. "Instead, binding properties such as the docking orientation is defined by regions of biophysical compatibility between these loops and the MHC surface."– The authors spend a lot of effort working through certain variables that contribute to the binding reaction. I am wondering if the authors took account of shape complementarity (e.g., PMID: 9628472) and CDR loops that carry different types of conserved canonical structures (e.g., PMID: 10656805). One could imagine that based on the protein folding requirements of CDR regions, certain residues are in the interface whereas others are internal to the CDR structure and cannot actually contribute directly to binding.

See response 5.

4. "We selected a key hydrogen bonding network in the KIR2DL2-HLA-C*07:02 interface [50] and compared this to the evolutionarily conserved YXY motif of CDR2β [19, 21]"– This is a good example to discuss the point above. it is important to know structurally, where each residue is. For example, the first Y (46 or 48 depending upon the nomenclature, above) often does not directly contribute to pMHC binding but may be important for the "outline structure" of the CDR loop itself. In addition, the authors do not discuss Van der Waals interactions really at all. Much of the TCR-pMHC interface (binding affinity) is driven by the exclusion of water, a property that is very difficult to assess on an amino acid-amino acid pairwise allotment of interaction energy. I was hoping that once the authors started to discuss "areas of binding potential" the contribution of non-side chain to side chain interactions would be discussed. It is unclear to this reviewer if these types of interactions are accounted for within their algorithm or if they are largely ignored.

See response 5.

Diving into this a bit more, this section was explicitly focused on debunking the concept of “conserved contact regions” where TCRs might be expected to frequently contact the same regions of the MHC molecule. The biophysical property analysis and sequence based analysis of Figure 3 suggests such rigidity in binding modes is unlikely. So, while one or the other Tyr may be involved in interacting with the MHC molecule, our main point is to say that we cannot know, a priori, precisely how these sidechains are arranged.

5. In discussing the interaction potential, of amino acids, the authors cite and discuss a single manuscript.42. P. Nandigrami, F. Szczepaniak, C.T. Boughter, F. Dehez, C. Chipot, and B. Roux. Computational assessment of protein-protein binding specificity within a family of synaptic surface receptors. Journal of Physical Chemistry B, 2022.There is of course an empirical and computational field of study for how proteins bind one another as well as for TCRs and pMHC (e.g., PMID: 10410805, PMID: 16193038, PMID: 18946038, PMID: 27348411). Some more inclusive discussion of past ideas about how proteins interact with one another and whether old ideas remain accurate could add to the overall discussion.

See response 1. We have also attempted to contextualize our results with previous studies in the discussion.

Regarding the citations listed, given the employment of the CHARMM force field in this newest version of the manuscript, we are aware of the exceptional literature focused on the empirical and computational study of protein binding. It is through our familiarity with this literature that we have sought to expand upon what has been done previously.

Again, in our eyes the ideal study would start entirely from crystal structures and simulate these structures using these empirically- and theoretically-derived force fields, and generate experimental predictions based upon these results. However, given the scale of the problem of interest of this manuscript, we need to reconsider how the problem is approached. We believe that the AIMS pseudo-structural approach is a complementary tool to these more explicit structural analyses.

6. "Figure 4: Interaction score between every TRBV (A) or TRAV (B) sequence and HLA allele for all four germline-encoded CDR loops. "– Why are alanine and glycines assumed to be zero/non-interacting? Does a binding reaction care if a contact is a side chain-side chain, backbone-backbone, or mix? Indeed, when the authors "counted the contacts" I assume many of the side chains were indeed interacting with backbone atoms. It has also been suggested that some side chains can contribute negatively to interfaces (e.g., PMID: 17041605). Another question, perhaps for the algorithm used, does it take into account the frequency at which say X and Y amino acids actually occur at a possible site of interactions.

See response 2.

Regarding the final question of amino acid frequency, the algorithm does not take this into account. The hope is that the AIMS interaction potential can provide unbiased estimates of CDR interaction propensities, allowing users to input mutated CDR loops or CDR/MHC sequences from other organisms.

7. It is mentioned that autoimmune-prone T cell repertoires are biased for certain TCR usage, does this bias include matching/non-matching HLA areas of recognition? There was some discussion on this but a clearer picture (if there is one) could be spelled out for the non-expert.

We thank the reviewer for highlighting another area where we can expand upon our thinking in the discussion regarding how our results tie into the greater picture of autoimmunity. Unfortunately, there are too few structures of TCRs with singular autoimmune antigens and TRAV/TRBV usage to make any broad claims about repeated regions of recognition. Further, our results actually predict that these autoimmune TCRs may not have matching contact regions on the HLA surface. Rather, the weaker interacting TRAV/TRBV alleles should encourage an increased dependence on CDR3 involvement.

Instead of adding further speculation regarding how our results may tie into the recognition of autoimmune pMHC complexes, we have added an extra paragraph to the discussion to provide context as to where our analysis is currently incapable of explaining certain patterns of autoimmunity:

“While over-reliance on CDR3 may bias TCRs towards auto- and cross-reactivity, the role of the MHC molecule in autoimmunity remains unclear from these results. In contexts such as celiac disease, a particular allele (HLA-DQ2.5) is enriched in patients with the disease [Qiao 2014, Gunnarsen 2017]. Given that the interaction potentials largely predict similar interaction strengths across HLA molecules (Figures 4, S8) such enrichment cannot yet be explained by AIMS. However, considering that the majority of diversity across HLA alleles is concentrated in the peptide binding regions (Figure 2), these correlations between HLA molecules and disease may largely be related to the peptides presented by these molecules, as has been suggested previously [Ishigaki 2022]. In trying to understand how allelic variation alters peptide presentation, and how this in turn impacts the onset of autoimmunity, the lack of strong rules for binding again complicates the problem. While HLA molecules have amino acid preferences at certain anchor positions, there are many exceptions to these "rules" [Nguyen 2021], and mutations to peptides that should improve stability in the HLA binding pocket can have unintended consequences on T cell activation [Smith 2021]. The substantial diversity of possible presented peptides makes the systematic analysis of this aspect of the autoimmunity problem exceptionally challenging.”

8. The interaction potentials also succeed in predicting TCR complexes that will not make contact with MHC. 20 of the 22 structures predicted to have poor CDR2β binding make no contact with MHC, while the last two only make one contact with MHC (Figure 5C). Further, all 8 structures predicted to have poor CDR1β binding make no contact with MHC. (Figure 5C). Again, this prediction accuracy is lower for class II predictions (Figure 5D).– This is a super interesting idea that may unlock a lot of what is going on. One wonders how much of this is random chance, i.e., if a different TCR-pMHC with the same V genes and HLA would behave similarly. Also, do these structures preclude (or are driven by) CDR1-peptide contacts, or are the structures carry such a different docking orientation as to completely preclude the CDR1 and CDR2 regions from being part of the binding interface?

We appreciate the excitement of the reviewer, but as reviewer 1 pointed out, our original methods were somewhat flawed. Our original definition of “contacts” was too stringent. However, with these new, more lenient definitions of contacts, we still find that AIMS-identified weak interacting V-gene encoded CDR loops still make fewer contacts than those that AIMS predicts to be strong interactors.

Looking at the Reviewer 1 Comment 2 Response, we can see an entire list of TCR-pMHC complexes that utilize the same V-gene. In some cases, there really is little involvement of the CDR1 and CDR2 loops in the interface. In others, we see that many of these germline “interactions” might be more properly classified as “contacts” that have limited energetic contributions to the interaction interface. So, the AIMS interaction potential may provide less of a prediction of where or how sidechain contacts will form, but more of a prediction of the relative contributions of these contacts. The docking orientations in many cases will likely be dominated by CDR3, especially in the case of weakly interacting V-gene encoded CDR loops.

9. The exposed residues on the α2-helix of HLA class I molecules are enriched in alanine and glycine relative to the α1- helix, which is highly unlikely to be involved in a specific, orientation-altering productive interaction.- In practice, it is this reviewer's understanding that there are exit contributions of Van der Waals interactions at these sites. Indeed, early ideas suggested that the diagonal area of pMHC (MHCa 61, MHCb73) used this divot for shape complementary purposes.

See response 3

10. It is important to note that these interaction potentials take an unbiased approach, calculating every possible interaction between TCR and MHC residues to produce this final score.– It was unclear if the authors mean position by position, or did they weigh whether a residue was actually surface exposed and capable of being part of the binding interface.

As discussed in lines 579-587 we take only surface exposed residues of the MHC and the CDR loops of TCRalpha and TCRbeta as defined by IMGT:

“As with all AIMS analyses, the first step is to encode each sequence into an AIMS compatible matrix. In this encoding each amino acid in structurally conserved regions is represented as a number 1-21, with zeros padding gaps between these structurally conserved features. The encoding is straightforward for the TCR sequences, with only the germline encoded regions of the CDR loops included for each gene. For the MHC encoding, only the structurally relevant amino acids are included for optimal alignment of each unique sequence. In this case, the structurally relevant amino acids of the class I (or class II) molecules were divided into three distinct groups; the TCR-exposed residues of the alpha1 (or α) helix, the TCR-exposed residues of the alpha2 (or β) helix, and the peptide-contacting residues of the given MHC.”

11. Productive side-chain interactions between CDR loops and the solvent-exposed residues of the MHC helices.– There does seem to be an (over) emphasis on side-chain interactions. And less so on the clustered ability for VDW and/or inhibitory interaction.

This is true, and is again due to our structure-free treatment of the interaction. As SCSC interactions are the most common in the interface, we believe this is a good first approximation to understanding how the germline regions of the TCR interact (see Response 5).

12. In general, there are quite a number of T cell development citations with actually very little discussed the role of thymic selection in and/or clonal T cell responses in skewing the TCR-pMHC interface to conform to selective pressures. E.g., TCRs can't be too good/cross-reactive or they would undergo central tolerance.

Unfortunately, given the datasets currently available to us for the analysis of the effects of selection and clonal expansion on repertoire skewing, any of our thoughts on this matter would be purely speculative in nature. We discuss the fantastic work done on the role of selection in shaping the docking angle solely to introduce the concept of “conserved contacts”. Our calculated interaction potentials are assumed to be antigen independent, and act as first approximations to the probable TCR-pMHC interface that is then likely altered by the antigen-CDR3 interaction. We briefly discuss these concepts in the results (Lines 292-304) and the Discussion (Lines 475-481).

13. "In calculating the interaction score, we assume that productive contacts are only made by the side chains of the interacting residues. This simplification does not capture all TCR-pMHC complex contacts, but here we are looking for selectivity enforced by specific TCR-MHC interactions mediated by side-chains."– Though stated as a caveat, perhaps some effort could be made to include side-chain to the backbone, etc interactions.

In addition to the incorporation of SC-backbone interactions in the parsing of deposited structures, we had initially also hoped to be able to include a backbone scoring function into the AIMS analysis. However, this appears to be beyond the scope of the current study, as the propensity for each sidechain to participate in backbone hydrogen bonding (either as the backbone partner or sidechain partner) is strongly determined by the distribution of the amino acids in either the CDRs or MHC α-helices of structures crystallized thus far.

The ideal backbone interaction scoring matrix would be unbiased, allowing for us to make estimates into the interactions between as yet uncrystallized TCR-MHC pairs. Currently, the Tyr-Ala and Tyr-Gln interactions would dominate the SC-backbone interaction scoring matrix, despite this potentially being a consequence of other, stronger interactions across the interface.

Reviewer #3 (Recommendations for the authors):Boughter and Meier-Schellersheim describe an analysis of TCRpeptide/MHC complexes, aiming to gain an understanding of the underpinnings of the "common" TCR binding geometry. This is fundamental to understanding the MHC restriction of TCRs and how T cells scan and readout peptides. They begin with a comprehensive bioinformatics approach, move to a structural analysis to help interpret the informatics, and bring in biophysical computations. The overall conclusion that specific contacts between TCR genes and MHC proteins are not necessarily preprogrammed and that traditional TCR binding geometries emerge from biophysical compatibility is supported by the data and consistent with recent findings. In general, the work and the conclusions are an advance and place recent findings into perspective. However, the strength of evidence is weakened by choices made in characterizing structures, computing energies, and a strained reliance on "roles" played the parts of the interface which have been discounted many times yet persist in the literature. The latter in particular weakens the discussion and how the authors view the impact of their work.The major strength of the paper is the approach taken; I found the comparative analysis of TCR and MHC genetic variability at the sequence level particularly compelling. Bringing in KIRs as a control was also a strong way to support the arguments. There is one major technical weakness in that, as far as is clear from the methods, interatomic interactions were considered with a 3.5 Å cutoff. This is woefully inadequate. Electrostatic interactions can be strong at long distances, which the authors really need to consider – say, going out to 6 Angstroms or so (there is much-published literature on short- and long-range electrostatics in protein interfaces). The importance of long-range electrostatics in TCR-peptide/MHC complexes has been demonstrated previously, particularly in prior work that aimed to address the same problem studied here. The authors also fall victim to the common immunology trope that CDR3-peptide interactions drive specificity, leaving CDR1/CDR2 to bind MHC proteins, i.e., the CDR loops have "roles" in binding. In the very first high-resolution structure of a TCR-peptide/MHC complex, CDR3 interactions with a class I MHC were noted and remarked on, as were CDR1 and CDR2 interactions with the peptide. Later work showed that these CDR3MHC and CDR2-peptide interactions were critical for binding. These findings have been replicated several times now. The authors' introduction of this perspective of different loops of the TCR playing evolved roles (CDR3->peptide, CDR1/2->MHC), and their interpretation of their findings in light of it, weakens the papers' conclusions and impact, and it is a missed opportunity that can be addressed with the authors' approach.The authors also should consider other literature for a greater impact on their work. For example, they also exclude backbone interactions – this is a curious omission from a biophysical perspective, and others in the field have published on the importance of backbone-mediated interactions (hydrogen bonds mostly) in stabilizing TCR interfaces. The authors also mention but fail to address T cell selection and the role of selection (and possibly coreceptor) in 'enforcing' what we get and have seen structurally (i.e., the idea that pre-selection TCRs bind all over the place, but selection ensures we get ones that bind right and work). Much has been written about this and it should be included.1) The very first high-resolution crystal structure of a TCR-pMHC complex by Garboczi and Wiley in the 90s (PMID 8906788) showed CDR3 contacts to the MHC and germline CDR1/2 contacts to the MHC. Later biophysical studies by our own group showed these were crucial for binding (PMID 23736024). Other work has shown the same. Thus, although it is common to say that diverse CDR3 loops bind peptide and germlineencoded CDR1/2 loops bind the MHC, this is not supported at the atomic or energetic level. It actually plays INTO the authors' argument about opportunism/compatibility, but curiously the authors do not discuss it. They should. These observations and the idea that "roles" are not hardcoded into the TCR CDR loops play right into the authors' opportunistic argument introduced at the end of the paper.

See response 5. Additionally, we have added this point about the “opportunism” of CDR1/2 binding peptide and CDR3 binding MHC to the discussion, elaborating further on our previous point about these loops needing to adapt to each individual target (as in the case of the super-bulged peptide)

2) A 3.5 Å cutoff is far too limited and ignores long-range electrostatics. Our own work addressing the same problem (which also introduced the notion of opportunism/compatibility) found signals for some "sloppy" evolved compatibility but only if we moved to longer ranges (PMID 26884163). The authors should re-evaluate their energetic analysis using longer-range cutoffs. To avoid greatly complicating the analysis, longer ranges could be done only with charged side chains. It was also very curious to omit main chain interactions, something which the authors might want to work back in (see PMID 17041605).

See response 3.

3) The authors should really address the question of how thymic education influences what we see. For example, we recently published a TCR that binds with an outlier geometry (not reverse) which signals just fine – an example of a class-mismatched TCR (emerged from a CD4^+^ T cell but binds a class I). This TCR is a bit weird in that it has an unusually long CDR3b loop that contacts both peptide and MHC (point 1 again). We also concluded that this is a weird TCR that somehow escaped normal thymic selection, implying that maybe the pre-selection repertoire has TCRs that bind crazily and one role of thymic selection is to filter these, giving us TCRs that are somehow "better" biologically (maybe they signal better, or possess lower x-reactivity, etc.). The authors need to work this thinking in. Relevant papers are PMID 36424374 and PMID 30833553.

At the time of depositing this preprint, your group’s very interesting class-mismatched TCR structure was not yet published. It does, however, fit nicely into our existing discussion of the reverse-docking TCRs and super-bulged peptide. It provides another, distinct example of non-canonical binding. Further speculation into how this fits into the greater picture of TCR-pMHC binding has now been added to the discussion lines 498508*:*

“New results further suggest the extent of the germline interaction permissiveness, with a class-mismatched CD4^+^ T cell capable of binding to and being activated by MHC class I, albeit with a slightly abnormal, but not reversed, docking orientation [Singh 2022]. These results further highlight the opportunism, expanding on previous work in this space [Blevins 2016], of TCR interactions in general, where the "rules" of TCR-pMHC binding seem to be more like guidelines. The literature has long focused on rules of interaction and commonalities between structures [Ysern 1998, al-lazikani 2000], which have been very helpful in guiding research over the past few decades. However, results such as the class-mismatched TCR, reversed docking TCRs, and super-bulged peptide suggest that perhaps such TCR-MHC specific rules may be too restrictive, and that these interactions may frequently involve more opportunistic configurations that call for unbiased evaluation.”

4) The authors use "compatibility" and "opportunistic" to describe TCR binding from a biophysical perspective, contrasting this with the hard-coded model. These are not new concepts though, and although the authors have greatly expanded on the topic (albeit with the limitations above), they should make note of this. They do reference some of the appropriate literature, but clarifying how they are expanding on the topic would strengthen the impact of the work.

We of course realize that this idea of broad biophysical compatibility and opportunism is not solely our own. We have expanded upon the previous discussion, and more explicitly described where our work expands on these previous ideas (see above excerpt from the text).

[Editors’ note: what follows is the authors’ response to the second round of review.]

The manuscript has been improved but there are some remaining issues that need to be addressed, as outlined below:Essential revisions:As you can see from the report, the reviewers appreciate the changes done for revision. After an extensive discussion, the overall consensus of the reviewers is that while the concept of evolved biophysical compatibility is possible and a potential solution to the question of how TCRs could be biased towards MHC proteins given the massive diversity in both receptor and ligand, it is a concept that is exceptionally difficult to demonstrate and the paper still has some wishful thinking. For this manuscript to move forward, we request that you tone down the paper, remove the claims highlighted by reviewer #1, and present the concept as an interesting possibility for which some evidence is offered but no solid proof (see report from reviewer #1 for details).

We thank the editor for this summary, and hope that our edits now more clearly illustrate the novel insights our analysis provides while pointing out that alternative interpretations are possible, as is always the case in research.

However, we also wish to note respectfully that several of the issues brought forth by reviewer # 1 must be refuted based on careful analysis of the agreement between our computational predictions and experimental structural analyses.

We also had a discussion with regards to the suggestion of reviewer #2 to perform a similar analysis on BCRs to verify that the signal is not spurious. We acknowledge that this might be beyond the scope of the current paper. However, if the authors chose to do this analysis, it can help solidify some of the claims.

We agree that such an analysis of BCR interactions would be very useful if BCRs represented a suitable negative control for the results presented here. Unfortunately, that is not the case, as we explain in our response to Reviewer #2.

Reviewer #1 (Recommendations for the authors):I recognize the time and effort that the authors have invested in responding to the reviews of the first version of the manuscript. It is appreciated that they recognized the circularity of the original Figure 5 and removed it, adjusted the distance thresholds and sequence-filters for contacts analysis, and that they have also removed references to the origin of self-reactive TCRs.

We thank Reviewer 1 for their initial comments, which have improved the quality of the manuscript.

My concerns with regard to the claims about V-gene interaction potential and determinants of the binding mode still stand, since the relevant text hasn't been modified and the author's responses are not convincing.

In their previous assessment, Reviewer 1 had raised issues regarding the “arbitrary” nature of the interaction potential, the details of the calculation of TCR-MHC interaction propensities, the aforementioned circularity of the original Figure 5, and then provided examples where our computational predictions might fall short of describing experimental observations.

In addition to the modifications we introduced in our previous response we have now edited the text where remaining issues were pointed out, changing language such as “validate” to “shows good agreement” and “exceptional agreement” to “good agreement” among other changes in the discussion. We hope these changes address the concerns of the reviewer and editor.

For example, the detailed analysis of the TRBV7-2 containing complexes provided by the authors in the response appears to disprove the AIMS-based prediction that this gene has low interaction potential: "Certainly PRESTO agrees with these structural interpretations, suggesting CDR2B dominates the germline interactions here, with 13/15 SC-SC contacts."

Here we do concede the selected text in the initial response to reviewers displays unclear writing. We meant “PRESTO agrees with the structural interpretations [of the reviewer]”. Given that PRESTO could be confused with the AIMS interaction scoring, we apologize for this lack of clarity. PRESTO is instead solely used for the automated counting of contacts in structures.

However, notwithstanding the initially unclear wording for this PDB entry, the remainder of our detailed analysis strongly supports our computational predictions. We could reiterate the analyses we provided for 5EU6 [Altered Self Antigen], 5D2L [CMV Antigen], 4MJI [HIV Antigen], and 40ZH [Celiac Autoimmune] but refer to the previous rebuttal instead, emphasizing that the conclusions we provided remain valid and appear to suggest the CDR loops encoded by TRBV7-2/7-3 have a propensity for limited or weak interactions with MHC.

The contorted logic that the authors produce to explain this disconnect doesn't really make sense: "However, yet again we have an abnormally high number of CDR2B backbone-backbone interactions, 14, suggestive of nonspecific tight packing not driven by TCRB specific interactions". What exactly is "nonspecific tight packing"?

We apologize if the notion of 'nonspecific tight packing’ was unclear in this context. We were using this term to describe regions of close contacts between two proteins that are formed not due to their own strong interactions, but instead due to distal interactions elsewhere on the TCR-pMHC interface**.**

We note that the authors of the PDB 4GRL study [PMID 24136005] highlight that CDR3A “dominate[s] the energetic landscape” of the interaction, supporting our interpretation of the predicted interaction propensity of TRBV7-2/7-3 and its agreement with experimental results.

In this case, CDR3A is the aforementioned “distal interaction” driving the complex formation, and the tight packing of CDR2B is merely a consequence of the docking, i.e. nonspecific. Briefly, CDR3A is the cause of the tight packing of CDR2B, and CDR2B in turn has no specific side chain interactions with the MHC. These claims are further quantitatively supported by Supplemental Table S5 of PMID 24136005 where mutagenesis of the β chain has only a modest impact on binding affinity, yet again suggestive of a limited interaction between the side chains of TRBV7-3 and the MHC α helix, further supporting the claims put forward by our analysis.

The authors also continue to over-sell their findings in the newly introduced text. For example, in describing the new Figure 5, the authors state: "This comparison shows exceptional agreement between our bioinformatic results and structural analyses". But when one compares Figure 5a and 5b, for example, the agreement is pretty dubious.

As noted before, we are now even more careful and have modified the text and replaced “exceptional agreement” by “good agreement”. However, we can quantify this agreement more rigorously by flattening the matrices of Figure 5a and 5b and comparing the distributions of contacts for each residue pair interaction potential score (Author response image 7).

**Author response image 7. sa2fig7:** 

We see that in the case of AIMS interactions scores greater than or equal to zero, we find the majority of our high frequency amino acid contacts. Note that the poor correlation (correlation coefficient 0.2) is expected for multiple reasons. First the TCR-pMHC crystal contact matrix is sparse, giving a high instance of “0” contact pairs. Further, hydrophobic amino acids are relatively rare both in the CDR loops and in the TCR contacting residues on the MHC helix. Lastly, “negative” interactions, such as Lys-Lys or Arg-Phe pairings, cannot have a corresponding negative contact count. Instead we see a higher proportion of 0 contacts between residue pairs.

And in 5d, *none* of the differences are significant, and many show the wrong directionality, for example, the median value for "Weak TRBV" is always greater than or equal to the median value for "Moderate TRBV".

Looking at Figure 5C/5D and Supplemental Figure S7, the data are consistent with the differences (or lack thereof) pointed out by us. Whereas the TRAV alleles identified as “strong” binders have much higher AIMS interaction scores than either the “moderate” (~1.5 unit difference) or the “weak” (~2 unit difference) binders, these interaction score differences are more modest for TRBV (~1 unit difference between “strong” and “weak” highest scores). This rationale is already highlighted in the text:

“Interestingly, these interaction potentials show better experimental agreement and predictive power for TRAV-encoded sequences compared to TRBV-encoded sequences. This could be due either to a fundamental difference in how TCRα and TCRβ contact the MHC α-helices, or in part due to the aforementioned higher interaction potential variance for TRAV-encoded CDR loops (Figure S7).”

And in the new text describing the AIMS potential: "The AIMS interaction potential, which can swiftly analyze thousands of sequences, has significantly outperformed more physically detailed and computationally expensive models. In a binary classification of a large database of structurally similar protein complexes, the AIMS interaction potential was capable of distinguishing binders and non-binders to an accuracy of 80%, whereas calculations run on over 45µs of simulated all atom trajectories could only distinguish to an accuracy of 50%. " I looked back at this reference, and what the authors neglect to mention is that the 80% performance comes from a highly parameterized model based on a linear discriminant analysis fitting a weight for each pair of residues in the interface-- it's not at all analogous to the calculation here in which AIMS scores are directly summed up. It's also a single family of interacting proteins.

We agree with the reviewer that the parametrization of the model plays an important role here. But we also wish to point out that in the reference in question [PMID 35787023] the more physically detailed and computationally expensive model used the same linear discriminant analysis-based approach on a similarly highly parametrized model, and was still substantially outperformed by the AIMS model (Figure 8: Computationally expensive method, Figure 9: AIMS model, in the referenced manuscript). Nonetheless, we have updated the text to highlight that the AIMS accuracies highlighted in the text do come from a linear discriminant-based analysis.

Further, this reference is directly analogous to the calculations presented in this manuscript, we just do not currently have a similar classification scheme in which to test TCR-pMHC binding. In fact, to our knowledge there exists no dataset where some number of CDR3-peptide pairings are kept constant while CDR1/2 are varied. This would provide the ideal test for the predictions included herein.

Lastly, while the Dpr-DIP interactome does concern a single family of interacting proteins, the TCR-pMHC interactome likewise concerns a single family of interacting proteins. Using a minimalist coarse graining of interactions between protein families with conserved structural features, we are able to provide reliable estimates of the tendencies for specific side chains to form productive interactions.

Reviewer #2 (Recommendations for the authors):With regards to the manuscript in general, in some places, the authors seem to want to have their cake and eat it too. Particularly, the idea that TCRs are evolutionarily biased to recognize MHC, included stating support for the "codon model" while at others suggesting that CDR1s and CDR2s have only minimal (complementary) roles in binding. With the extension suggesting that some TRAVs and TRBVs have no (or very minimal) MHC/HLA binding potential.

We agree with reviewer 2 that our conclusions could be more directed, and hope that the added text in the discussion reflects this. Specifically, we have added the point:

“Further, while the interaction codon hypothesis suggests co-evolved interaction interfaces between the TCR and MHC, our analysis instead suggests that each TRAV-TRBV pairing finds a unique approach to binding within the constraints permitted by the MHC molecular surface. In other words, the MHC molecule largely defines the interface.”

We note that we do not suggest that TCRs are evolutionarily biased to recognize MHC anywhere in the text. While we say our results are somewhat in agreement with the “codon model”, we note a “broader, dynamic interpretation” of the interaction is suggested by AIMS. In other words, there is no evolutionary bias. The “biophysical compatibility” we suggest is a much weaker assumption than an explicit evolutionary bias at the sequence level.

We wish to point out that, while reviewer 2 presents the codon model and a model with minimally interacting CDR 1/2 loops as models in opposition, both models are compatible with the data presented in this manuscript.

Specifically, we do not disregard or attempt to disprove the previous finding of evolutionary TCR-pMHC interactions involving TRBV8-2. Instead, we suggest that such conserved interactions are rare, and that some TRAV/TRBV alleles may exist that show very different tendencies, with a limited propensity for binding MHC. Indeed, this diversification of binding strategies may be evolutionarily advantageous, giving TCRs a range of possible strategies for recognizing antigen. In other words, given the large number of possible TRAV and TRBV combinations, TCRs can realize a multitude of distinct binding strategies.

This later argument would suggest that antibodies, fully capable of creating diverse CDR3s, should similarly have a (modest, strong) ability to bind pMHC ligands. I suppose a computational test of the general idea the authors are putting forward would be to use their AIMs platform with human antibody CDR1s and CDR2s to see if these were all net no-binding or negative binding with MHC. However, I do not like the idea of bringing up additional questions/tests of the model during a re-review.

We appreciate the reviewer’s concern with our workload in this re-review. However, we have already considered antibody CDR loops as a negative control, and were surprised to find that they represent a poor negative control. Why we don’t see more natural antibody-pMHC interactions seems to be an immunological question, rather than a biophysical one.

A relatively recent review [PMID 31544838] highlights 52 previously published “TCR-like” antibodies binding to class I MHC molecules with peptide specificities, with all measured affinities in the nanomolar range. The authors of this review note that despite excellent synthetic CDR loop libraries, most TCR-like antibodies are “isolated from libraries built on endogenous variable gene repertoires”. In other words, using native IGHV/IGLV/IGKV libraries does not preclude these antibodies from binding to pMHC. While some (or perhaps the majority) of these “TCR-like” antibodies do not adopt this canonical TCR-pMHC docking orientation, at least one study [PMID 19307587] finds a “TCR-like” antibody that does. Importantly, while the heavy chain overlaps with the TCR-α interface and the light chain overlaps with the TCR-β interface, the antibody germline CDR loops bear little to no resemblance to either chain (Author response table 2).

**Author response table 1. sa2table1:** Comparison of CDR loop sequences of TCR 1G4 and the TCR-like antibody that binds to the same pMHC complex. Sequences identified via PMID 19307587

Loop	CDR1A/H	CDR2A/H	CDR1B/L	CDR2B/L
TCR Seq	DSAIYN	IQSSQRE	MNHEY	SVGAGI
Antibody Seq	GFTFSTYQ	IVSSGGST	TGTSRDVGGYNYVS	DVIERSS

As such, the latter argument outlined by reviewer 2 does seem, at least in part, to be consistent for antibodies, and further they do not represent a suitable negative control.